

# Evaluation of Arctic warming in mid-Pliocene climate simulations

Wesley de Nooijer[1], Qiong Zhang[1], Qiang Li[1], Qiang Zhang[1], Xiangyu Li[2,3], Zhongshi Zhang[4,3,2], Chuncheng Guo[3], Kerim H. Nisancioglu[3], Alan M. Haywood[5], Julia C. Tindall[5], Stephen J. Hunter[5], Harry J. Dowsett[6], Christian Stepanek[7], Gerrit Lohmann[7], Bette L. Otto-Bliesner[8], Ran Feng[9], Linda E. Sohl[10], Ning Tan[11,12], Camille Contoux[12], Gilles Ramstein[12], Michiel L. J. Baatsen[13], Anna S. von der Heydt[13,14], Deepak Chandan[15], W. Richard Peltier[15], Ayako Abe-Ouchi[16], Wing-Le Chan[16], Youichi Kamae[17], Chris M. Brierley[18]

1. Department of Physical Geography and Bolin Centre for Climate Research, Stockholm University, Stockholm, Sweden
2. Institute of Atmospheric Physics, Chinese Academy of Sciences, Beijing, China
3. NORCE Norwegian Research Centre, Bjerknes Centre for Climate Research, Bergen, Norway
4. Department of Atmospheric Science, School of Environmental Studies, China University of Geosciences, Wuhan, China
5. School of Earth and Environment, University of Leeds, Woodhouse Lane, Leeds, West Yorkshire, UK
6. Florence Bascom Geoscience Center, U.S. Geological Survey, Reston, VA 20192, USA
7. Alfred Wegener Institute - Helmholtz-Zentrum für Polar und Meeresforschung, Bremerhaven, Germany
8. Palaeo and Polar Climate Division, National Center for Atmospheric Research, Boulder, Colorado, USA
9. Department of Geosciences, College of Liberal Arts and Sciences, University of Connecticut, Connecticut, USA
10. CCSR/GISS, Columbia University, New York, USA
11. Key Laboratory of Cenozoic Geology and Environment, Institute of Geology and Geophysics, Chinese Academy of Sciences, Beijing, China
12. Laboratoire des Sciences du Climat et de l'Environnement, LSCE/IPSL, CEA-CNRS-UVSQ, Université Paris-Saclay, Gif-sur-Yvette, France
13. Centre for Complex Systems Science, Utrecht University, Utrecht, The Netherlands
14. Institute for Marine and Atmospheric research Utrecht (IMAU), Department of Physics, Utrecht University, Utrecht, The Netherlands.
15. Department of Physics, University of Toronto, Toronto, Ontario, Canada
16. Centre for Earth Surface System Dynamics (CESD), Atmosphere and Ocean Research Institute (AORI), University of Tokyo, Tokyo, Japan
17. Faculty of Life and Environmental Sciences, University of Tsukuba, Tsukuba, Japan
18. Department of Geography, University College London, London, UK

*Correspondence to*: Qiong Zhang (qiong.zhang@natgeo.su.se)

**Abstract.** Palaeoclimate simulations improve our understanding of the climate, inform us about the performance of climate models in a different climate scenario, and help to identify robust features of the climate system. Here, we analyse Arctic warming in an ensemble of 16 simulations of the mid-Pliocene Warm Period (mPWP), derived from the Pliocene Model Intercomparison Project Phase 2 (PlioMIP2).

The PlioMIP2 ensemble simulates Arctic (60-90° N) annual mean surface air temperature (SAT) increases of 3.7 to 11.6 °C compared to the pre-industrial, with a multi-model mean (MMM) increase of 7.2 °C. The Arctic warming amplification ratio relative to global SAT anomalies in the ensemble ranges from 1.8 to 3.1 (MMM is 2.3). Sea ice extent anomalies range from



-3.0 to -10.4 x $10^6$ km$^2$ with a MMM anomaly of -5.6 x$10^6$ km$^2$, which constitutes a decrease of 53 % compared to the pre-industrial. The majority (11 out of 16) models simulate summer sea ice-free conditions ($\leq$ 1 x $10^6$ km$^2$) in their mPWP simulation. The ensemble tends to underestimate SAT in the Arctic when compared to available reconstructions. The simulations with the highest Arctic SAT anomalies tend to match the proxy dataset in its current form better. The ensemble shows some agreement with reconstructions of sea ice, particularly with regards to seasonal sea ice. Large uncertainties limit the confidence that can be placed in the findings and the compatibility of the different proxy datasets. We show that, while reducing uncertainties in the reconstructions could decrease the SAT data-model discord substantially, further improvements are likely to be found in enhanced boundary conditions or model physics. Lastly, we compare the Arctic warming in the mPWP to projections of future Arctic warming and find that the PlioMIP2 ensemble simulates greater Arctic amplification, an increase instead of a decrease in AMOC strength compared to pre-industrial, and a lesser strengthening of northern modes of variability than CMIP5 future climate simulations. The results highlight the importance of slow feedbacks in equilibrium climate simulations, and that caution must be taken when using simulations of the mPWP as an analogue for future climate change.

## 1 Introduction

The simulation of past climates improves our understanding of the climate system, and it provides an opportunity for the evaluation of the performance of climate models beyond the range of present and recent climate variability (Braconnot et al., 2012; Harrison et al., 2014, 2015; Masson-Delmotte et al., 2013; Schmidt et al., 2014). Comparisons of palaeoclimate simulations and palaeoenvironmental reconstructions have been carried out for several decades (Braconnot et al., 2007; Joussaume and Taylor, 1995) and show that while climate models can reproduce the direction and large-scale patterns of changes in climate, they tend to underestimate the magnitude of specific changes in regional climates (Braconnot et al., 2012; Harrison et al., 2015). The comparison of palaeoclimate simulations with future projections has aided in the identification of robust features of the climate system which can help constrain future projections (Harrison et al., 2015; Schmidt et al., 2014), including in the Arctic (Yoshimori and Suzuki, 2019).

One such robust feature is the Arctic amplification of global temperature anomalies (Serreze and Barry, 2011). Increased warming in the Arctic region compared to the global average is a common feature of both palaeo- and future climate simulations and is also present in the observational record (Collins et al., 2013; Masson-Delmotte et al., 2013). Arctic warming has a distinct seasonal character, with the largest sea surface temperature (SST) and the smallest surface air temperature (SAT) anomalies occurring in the summer due to enhanced ocean heat uptake following sea ice melt (Serreze et al., 2009; Zheng et al., 2019). It is critical to correctly simulate Arctic amplification as future warming in the Arctic directly affects ice sheet stability, global sea-level rise and carbon cycle feedbacks (e.g. through permafrost melting; Masson-Delmotte et al., 2013). Several multi-model analyses that included palaeoclimate simulations and/or future projections found



that changes in northern high-latitude temperatures scale (roughly) linearly with changes in global temperatures (Bracegirdle
and Stephenson, 2013; Harrison et al., 2015; Izumi et al., 2013; Masson-Delmotte et al., 2006; Miller et al., 2010; Schmidt et
al., 2014; Winton, 2008).

Underestimation of Arctic SAT has been reported for several climates in the Palaeoclimate Modelling Intercomparison
Project Phase 3 (PMIP3), including the mid-Pliocene Warm Period (Dowsett et al., 2012; Haywood et al., 2013a; Salzmann
et al., 2013), Last Interglacial (LIG: Bakker et al., 2012; Lunt et al., 2013; Otto-Bliesner et al., 2013) and Eocene (Lunt et al.,
2012a). PMIP4 simulations, however, of the LIG showed good agreement with SAT reconstructions in the Canadian Arctic,
Greenland, and Scandinavia, while showing overestimations in other regions (Otto-Bliesner et al., 2020). PMIP4 simulations
of the Eocene were also able to capture the polar amplification indicated by SAT proxies (Lunt et al., 2020).

In the present work, we analyze the simulated Arctic warming in a new ensemble of 16 simulations in the Pliocene Model
Intercomparison Project Phase 2 (PlioMIP2) (Haywood et al., 2016). PlioMIP2 is designed to represent a discrete time slice
within the mid-Pliocene Warm Period (mPWP; 3.264–3.025 Ma; sometimes referred to as mid-Piacenzian Warm Period):
Marine Isotope Stage (MIS) KM5c, 3.204–3.207 Ma (Dowsett et al., 2016, 2013; Haywood et al., 2013b, 2016). The mPWP
is the most recent period in geological history with atmospheric $CO_2$ concentrations similar to the present, therefore
providing great potential to learn about warm climate states. This gives lessons learned from the mPWP potential relevance
for future climate change (Burke et al., 2018; Tierney et al., 2019), and this is one of the guiding principles of PlioMIP
(Haywood et al., 2016).

Palaeoenvironmental reconstructions show that the elevated $CO_2$ concentrations in the mPWP coincided with substantial
warming, which was particularly prominent in the Arctic (Brigham-Grette et al., 2013; Dowsett et al., 2012; Panitz et al.,
2016; Salzmann et al., 2013). Haywood et al. (2020) discuss the large-scale outcomes of PlioMIP2 and observe a global
warming that is between the best estimates of predicted end-of-century global temperature change under the RCP6.0 (+2.2 ±
0.5 °C) and RCP8.5 (3.7 ± 0.5 °C; Collins et al., 2013) emission scenarios.

The dominant mechanism for global warming in mid-Pliocene simulations is through changes in radiative forcing following
increases in greenhouse gas concentrations (Chandan and Peltier, 2017; Hill et al., 2014; Hunter et al., 2019; Kamae et al.,
2016; Lunt et al., 2012b; Stepanek et al., 2020; Tan et al., 2020). Polar warming is also dominated by changes in greenhouse
gas emissivity (Hill et al., 2014; Tindall and Haywood, 2020). Apart from the changes in greenhouse gas concentrations,
changes in boundary conditions that led to warming in previous simulations of the mPWP included the specified ice sheets,
orography, and vegetation (Hill, 2015; Lunt et al., 2012b).




In PlioMIP1, the previous phase of this project, model simulations underestimated the strong Arctic warming that is inferred from proxy records was found (Dowsett et al., 2012; Haywood et al., 2013a; Salzmann et al., 2013). This data-model discord may have been caused by uncertainties in model physics, boundary conditions, or reconstructions (Haywood et al., 2013a)

Uncertainties in model physics include unconstrained forcings and uncertainties in model parameters. It was found that the inclusion of chemistry-climate feedbacks from vegetation and wildfire changes leads to substantial global warming (Unger and Yue, 2014, while excluding industrial pollutants and explicitly simulating aerosol-cloud interactions (Feng et al., 2019), and decreasing atmospheric dust loading (Sagoo and Storelvmo, 2017) leads to increased Arctic warming in mPWP simulations. Similarly, in simulations of the Eocene, two models that implemented modified aerosols had better skill than
other models at representing polar amplification (Lunt et al., 2020). Changes in model parameters, such as the sea ice albedo parameter (Howell et al., 2016b), may provide further opportunities for increasing data-model agreement in the Arctic.

Several studies found changes in boundary conditions that could help resolve some of the data-model discord in the Arctic for PlioMIP1 simulations. The studied changes in boundary conditions include changes in orbital forcing (Feng et al., 2017;
Prescott et al., 2014; Salzmann et al., 2013), atmospheric $CO_2$ concentrations (Feng et al., 2017; Howell et al., 2016b; Salzmann et al., 2013), and palaeogeography and bathymetry (Brierley and Fedorov, 2016; Feng et al., 2017; Hill, 2015; Otto-Bliesner et al., 2017; Robinson et al., 2011).

New in the experimental design of PlioMIP2 are a closed Bering Strait and Canadian Archipelago. The closure of these
Arctic Ocean gateways has been shown to alter oceanic heat transport into the North Atlantic (Brierley and Fedorov, 2016; Feng et al., 2017; Otto-Bliesner et al., 2017). Additionally, the focus on a specific time slice within the mPWP allows for reduced uncertainties in reconstructions and boundary conditions, in particular with regards to orbital forcing. These changes have led to an improved data-model agreement for reconstructions of SST, particularly in the North Atlantic (Dowsett et al., 2019; Haywood et al., 2020; McClymont et al., 2020). Multi-model mean (MMM) SST anomalies in the North Atlantic
deviate less than 3 °C from reconstructed temperatures (Haywood et al., 2020).

In the following sections, we first evaluate the simulated Arctic (60–90° N) temperatures and sea ice extents (SIE) in the PlioMIP2 ensemble. We then perform a data-model comparison for SAT and an evaluation of how uncertainties in the reconstructions may affect the outcomes of the data-model comparison. We then compare the simulated sea ice to
reconstructions. Lastly, we investigate some climatic features of the mPWP, including Arctic amplification, the Atlantic Meridional Overturning Circulation (AMOC) and northern modes of variability, and compare these analyses to findings of future climate studies to investigate the extent to which the mPWP can be used as an analogue for future Arctic climate change.



## 2 Methods

### 2.1 Participating models

The simulations of the mPWP by 16 models participating in PlioMIP2 were used in this study. The models included in this study are listed in Table 1. A more detailed description of each model's information and experiment setup can be found in Haywood et al. (2020).

For each simulation, the last 100 years of data are used for the analysis. Individual model resultsaere calculated on the native grid of each model. MMM results are obtained after regridding each model's output to a 2° x 2° grid using bilinear interpolation. Using a non-weighted ensemble mean theoretically averages out biases in models, assuming models are independent, and errors are random (Knutti et al., 2010). Climate models can, however, generally not be assumed to be independent (Knutti et al., 2010; Tebaldi and Knutti, 2007) and this is especially true for the PlioMIP2 ensemble where 150 many models have common origins (Table 1). The MMM results will therefore likely be biased towards specific common errors within the models comprising the ensemble.

**Table 1: Models participating in PlioMIP2 used in this study.**

| Model name | Institution | PlioMIP2 reference |
| --- | --- | --- |
| CCSM4-NCAR | National Center for Atmospheric Research (NCAR) | Feng et al. (in prep.) |
| CCSM4-Utrecht | IMAU, Utrecht University | |
| CCSM4-UofT | University of Toronto, Canada | Chandan and Peltier (2017) |
| CESM1.2 | NCAR | Feng et al. (in prep.) |
| CESM2 | NCAR | Feng et al. (in prep.) |
| COSMOS | Alfred Wegener Institute | Samakinwa et al., (2020); Stepanek et al. (2020) |
| EC-Earth 3.3 | Stockholm University | Zhang et al. (in prep.) |
| GISS−E2−1−G | NASA/GISS | Chandler et al. (in prep.) |
| HadCM3 | Hadley Centre for Climate Prediction and Research/Met Office UK | Hunter et al. (2019) |
| IPSLCM5A | Laboratoire des Sciences du Climat et de l'Environnement (LSCE) | Tan et al. (2020) |
| IPSLCM5A−2.1 | LSCE | Tan et al. (2020) |
| IPSL-CM6A−LR | LSCE | Contoux et al. (in prep.) |
| MIROC4m | CCSR/NIES/FRCGC, Japan | Chan et al. (in prep.) |





| MRI–CGCM2.3 | Meteorological Research Institute | Kamae et al. (2016) |
|---|---|---|
| NorESM-L | NORCE Norwegian Research Centre, Bjerknes Centre for Climate Research, Bergen, Norway | Li et al. (2020) |
| NorESM1-F | NORCE Norwegian Research Centre, Bjerknes Centre for Climate Research, Bergen, Norway | Li et al. (2020) |

## 2.2 Data-model comparisons

To evaluate the ability of climate models to simulate mPWP Arctic warming, we first perform a comparison to SAT estimates from palaeobotanical reconstructions. The data-model comparison is performed using temperature anomalies, calculated by differencing the mPWP and the pre-industrial simulation, to avoid overestimations of agreement due to strong latitudinal effects on temperature (Haywood and Valdes, 2004).

Reconstructed mPWP SATs are taken from Feng et al. (2017), who updated and combined compilations made by Ballantyne et al. (2010) and Salzmann et al. (2013) (Table S1). Qualitative estimates of confidence levels for each reconstruction were made by Feng et al. (2017) and Salzmann et al. (2013). Only reconstructions that are located at or northward of 60° N and for which the temporal range covers the KM5c time slice are included in the data-model comparison. Three reconstructions from Ballantyne et al. (2010) at the same location (78.3° N, -80.2° E) were averaged to avoid oversampling that location.


The data-model comparison will be a point-to-point comparison of modelled and reconstructed temperatures estimated from palaeobotanical proxies, which initially does not take the uncertainties of the reconstructions (Table S1) into account. The potential influence of the uncertainties in reconstructions on the outcomes of the data-model comparison will be investigated in a later section. The temporal range of the reconstructions is broad and certainly not resolved to the resolution of the KM5c
time slice, unlike the dataset of SST estimates compiled by Foley and Dowsett (2019) used for PlioMIP2 SST data-model comparisons by Haywood et al. (2020) and McClymont et al. (2020). Prescott et al. (2014) found that peak warmth in the mPWP would be diachronous between different regions based on simulations with different configurations of orbital forcing. Orbital forcing is particularly important in the high latitudes and for proxies that may record seasonal signatures (e.g. due to recording growing season temperatures). As such, there may be significant biases in the dataset, as the temporal ranges of the
proxies include periods with substantially different external forcing than during the KM5c time slice for which the simulations are run. Further uncertainties arise due to bioclimatic ranges of fossil assemblages, errors in pre-industrial temperatures from the observational record, and additional unquantifiable factors. Ultimately, the uncertainties constrain our ability to evaluate the Arctic warming in the PlioMIP2 simulations substantially. A more detailed description of the uncertainties in the SAT estimates can be found in the work of Salzmann et al. (2013).




The reconstructed temperatures are differenced with temperatures from the observational record to obtain proxy temperature anomalies. Observational record temperatures are obtained from the Berkeley Earth monthly land and ocean dataset (Rohde et al., 2013a, 2013b), and the average temperature in the 1870–1899 period was used.

Furthermore, the simulation of mPWP SIE will be evaluated using three palaeoenvironmental reconstructions that indicate whether sea ice was perennial or seasonal at a specific location. Darby (2008) infers that perennial sea ice was present at Lomonosov Ridge (87.5° N, 138.3° W) throughout the last 14 Ma based on estimates of drift rates of sea ice combined with inferred circum-Arctic sources of detrital mineral grains in sediments at this location. Knies et al. (2014) infer seasonal sea ice cover based on the abundance of the $IP_{25}$ biomarker, a lipid that is produced by certain sea ice diatoms, which is similar

to the modern summer minimum throughout the mid-Pliocene in sediments at two locations near the Fram Strait, of which one is chosen for this data-model comparison (80.2° N, 6.4° E). Similarly, Clotten et al. (2018) infer seasonal sea ice cover with occasional sea ice-free conditions in the Iceland Sea (69.1° N, -12.4° E) between 3.5 and 3.0 Ma using a multiproxy approach. As the sediment record studied by Clotten et al. (2018) included a peak in the abundance of the $IP_{25}$ biomarker at 3.2 Ma, we infer seasonal sea ice cover during the KM5c time slice.



**3 Arctic warming in the PlioMIP2 ensemble**

**3.1 Annual mean warming**



**Figure 1: Simulated global and Arctic (a) SAT anomalies (mPWP minus pre-industrial), (b) Arctic amplification ratio of SAT, and (c) SST anomalies for each model and the MMMs. The horizontal lines represent PlioMIP2 MMM values.**





The PlioMIP2 experiments show substantial increases in global annual mean SAT (ranging from 1.7 to 5.2 °C, with a MMM
of 3.2 °C; Fig. 1a; Table S2) and SST (ranging from 0.8 to 3.9 °C, with a MMM of 2.0 °C; Fig. 1c; Table S2) in the mPWP,
compared to pre-industrial. All models show a clear Arctic amplification, with annual mean SAT in the Arctic (60–90° N)
increasing by 3.7 to 11.6 °C (MMM of 7.2 °C; Fig. 1a). Annual mean SST in the Arctic increased by 1.3 to 4.6 °C (MMM of
2.4 °C; Fig. 1c). The magnitude of Arctic amplification, defined as the ratio between the Arctic and global SAT anomaly,
ranges from 1.8 to 3.1, and the MMM shows an Arctic amplification factor of 2.3 (Fig. 1b). Temperature anomalies in the
PlioMIP2 ensemble are similar but slightly higher, than in the PlioMIP1 ensemble. A similar magnitude of Arctic
amplification is simulated by the two ensemble means.

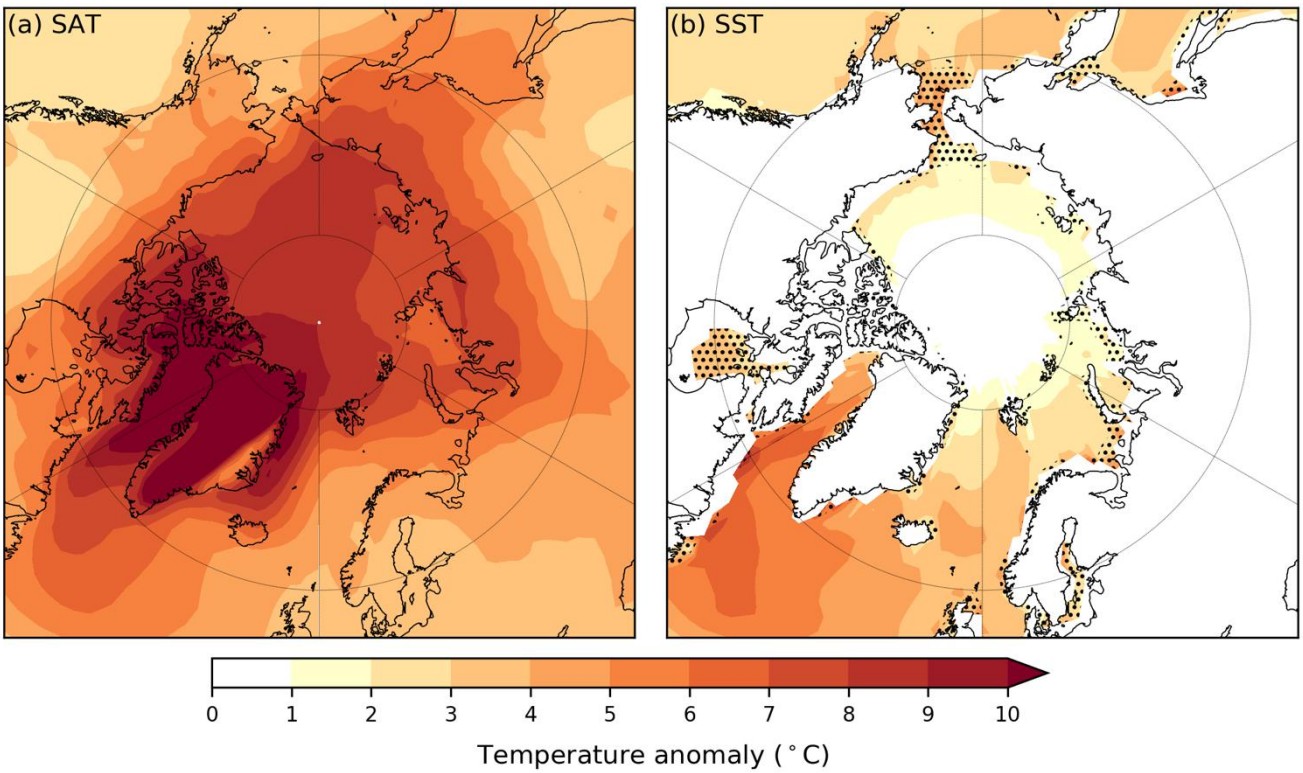

**Figure 2: MMM annual temperature anomalies in the Arctic: (a) SAT, (b) SST. At least 15 out of 16 models agree on the sign of**
**change at each location. Stippling in (b) denotes locations where five or less models provided data and where SSTs thus represent**
**only a subset of the ensemble. Differences between simulations in spatial extent of SSTs may arise due to different model**
**resolutions or implementation of the land-sea mask.**

The greatest MMM SAT anomalies in the Arctic coincide with local prescribed changes in boundary conditions (Fig. 2a).
The regions with reduced ice sheet extent on Greenland (Haywood et al., 2016) generally show warming of over 10 °C and
even up to 20°C. Additionally, temperature anomalies of over 10 °C are simulated around the Canadian Archipelago, which
is closed in the mPWP simulations (Haywood et al., 2016). SAT anomalies of around 6–9 °C are simulated over most of the



Arctic Ocean regions. SST anomalies in the Arctic are strongest in the Baffin Bay and the Labrador Sea, reaching up to 7 °C (Fig. 2b).

**3.2 Seasonal warming**

The distinct seasonality of Arctic amplification (Serreze et al., 2009; Zheng et al., 2019) can be used to identify mechanisms causing Arctic amplification. Figure 3 depicts the seasonality of Arctic warming for each model, with monthly SAT and SST anomalies normalized by the annual mean anomaly for that specific model.

The ensemble simulates a consistent peak in Arctic SST warming between July and September (Fig. 3b). This is consistent
with the response that increased seasonal heat storage from incoming heat fluxes would have upon the reduction of SIE (Serreze et al., 2009; Zheng et al., 2019). Minimum SAT warming is expected in the summer because of the increased ocean heat uptake, while maximum SAT warming is expected in the autumn and winter following the release of this heat (Pithan and Mauritsen, 2014; Serreze et al., 2009; Yoshimori and Suzuki, 2019; Zheng et al., 2019). This is not simulated by all models, however (Fig. 3a). COSMOS, GISS-E2-1-G, IPSL-CM6A-LR, and MRI-CGCM2.3 all do show this autumn and
winter amplification of annual mean SAT anomalies and decreased warming in the summer. Decreased summer warming is simulated by CCSM4-Utrecht, EC-Earth 3.3 and IPSLCM5A in combination with autumn amplification, and by CESM2 and NorESM1-F in combination with winter amplification. All other models in the ensemble do not show an autumn or winter amplification in combination with decreased summer warming, suggesting a more limited role of reductions in SIE underlying the seasonal cycle of Arctic SAT anomalies.

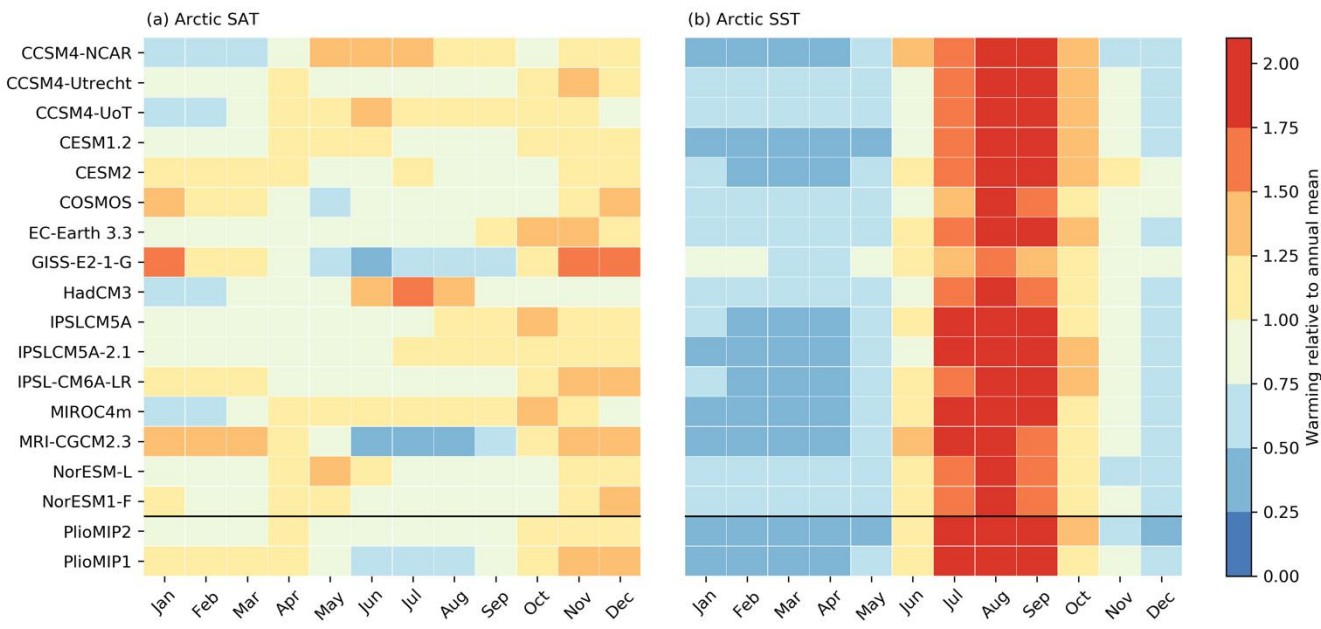




**Figure 3: Monthly mean Arctic (a) SAT and (b) SST warming relative to the respective annual mean Arctic warming for each model in the PlioMIP2 ensemble. Values of zero would imply no warming compared to pre-industrial in a given month.**

## 4 Sea ice analysis

### 4.1 Annual mean sea ice extent

The MMM of Arctic annual SIE (sea ice concentration $\geq$ 0.15) is 11.9 x $10^6$ km$^2$ for the pre-industrial simulations, and 5.6 x $10^6$ km$^2$ (a 53 % decrease) for the mPWP simulations. The pre-industrial annual mean SIE ranges from 9.1 to 15.6 x $10^6$ km$^2$ in the ensemble, while the mPWP SIE ranges from 2.3 to 10.4 x $10^6$ km$^2$. The decrease in SIE between individual simulations ranges from -3.0 x $10^6$ km$^2$ to -10.4 x $10^6$ km$^2$ (Table S2). Interestingly, the PlioMIP1 MMM shows larger SIEs in both the pre-industrial and the mPWP than any individual model in the PlioMIP2 ensemble (Fig. 4). The 53% MMM

decrease in SIE simulated by the PlioMIP2 ensemble is substantially greater than the 33% MMM decrease in SIE simulated by the PlioMIP1 ensemble (Howell et al., 2016a).

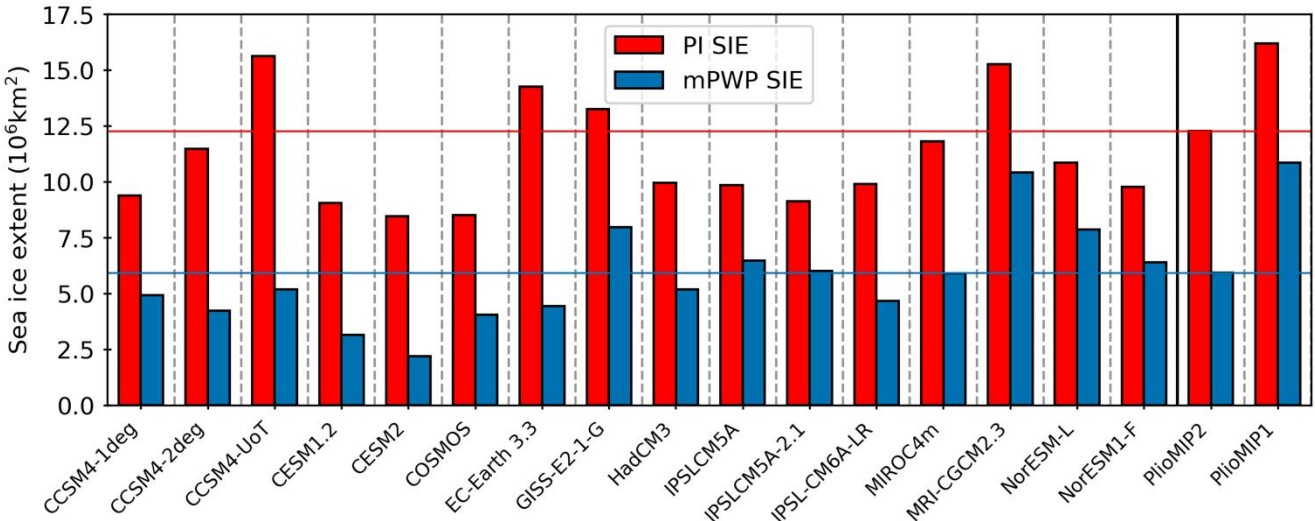

**Figure 4: Mean annual SIE ($10^6$ km$^2$) for the pre-industrial and mPWP simulations. The horizontal lines represent PlioMIP2 MMM values.**

**4.2 Monthly mean sea ice extent**

The seasonal cycle of SIE anomalies is depicted in Fig. 5a. Reductions in SIE are slightly greater in the autumn (September-November) as compared to other seasons for the MMM. There is, however, no consistent response in the seasonal character of SIE anomalies in the PlioMIP2 ensemble. CCSM4-UoT, CESM2, IPSLCM5A, IPSLCM5A-2.1 simulate the largest reductions in SIE in winter (December-February), while GISS-E2-1-G and HadCM3 simulate the largest SIE reductions in

spring. The remaining 10 models simulate the greatest SIE anomalies in autumn.



A more consistent response is observed when comparing monthly mean mPWP SIEs and pre-industrial SIEs. For each model, the largest reductions in SIE in terms of percentages occur between August and October (Fig. 5b). This may be explained by the lesser amount of energy that is needed to melt a given % of the smaller SIE that is present in the summer compared to winter. 11 out of 16 models simulate sea ice-free conditions (SIE < 1x10$^6$ km$^2$) in at least one month, while five models (GISS-E2-1-G, IPSLCM5A, IPSLCM5A-2.1, MRI-CGCM2.3, and NorESM-L) do not (Fig. 5b). The NorESM1-F simulation simulates the smallest global mean warming (1.7 °C; Fig. 1a) resulting in Arctic sea ice-free conditions.

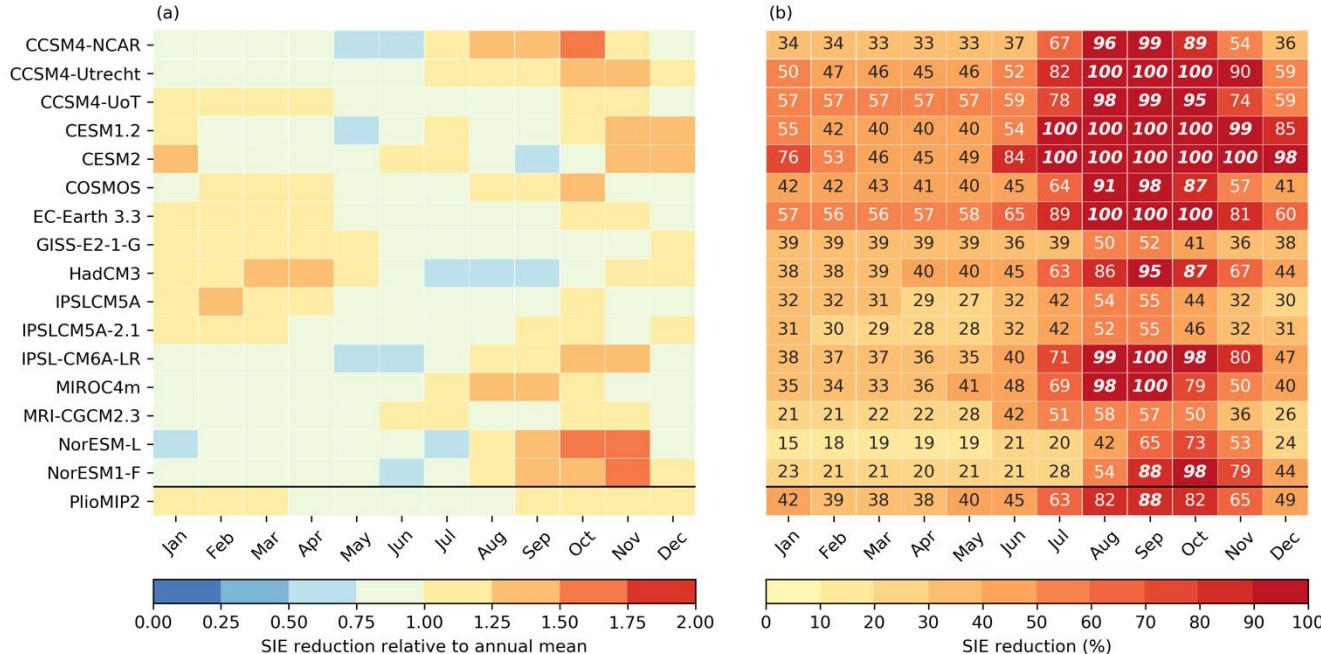

**Figure 5: (a) Monthly SIE anomalies relative to annual mean anomalies, warmer colours highlight in which months reductions in sea ice were largest. (b) Reduction in SIE (%) in the mPWP simulations compared to the pre-industrial monthly mean SIE for each month. Highlighted in bold italics in (b) are months with sea ice-free conditions (SIE < 1x10$^6$ km$^2$).**

### 4.3 Sea ice and Arctic warming

There is a strong anti-correlation between annual mean Arctic SAT and SIE anomalies (R=-0.79; Fig. 6a), as well as between SST and SIE anomalies (R=-0.79; Fig. 6b). These anti-correlations are stronger than those found for the PlioMIP1 ensemble (R=-0.76, R=-0.73, respectively; Howell et al., 2016). The results highlight the importance of correctly simulating SIE anomalies for the Arctic SAT anomalies, which, in turn, may yield better agreement with reconstructions.




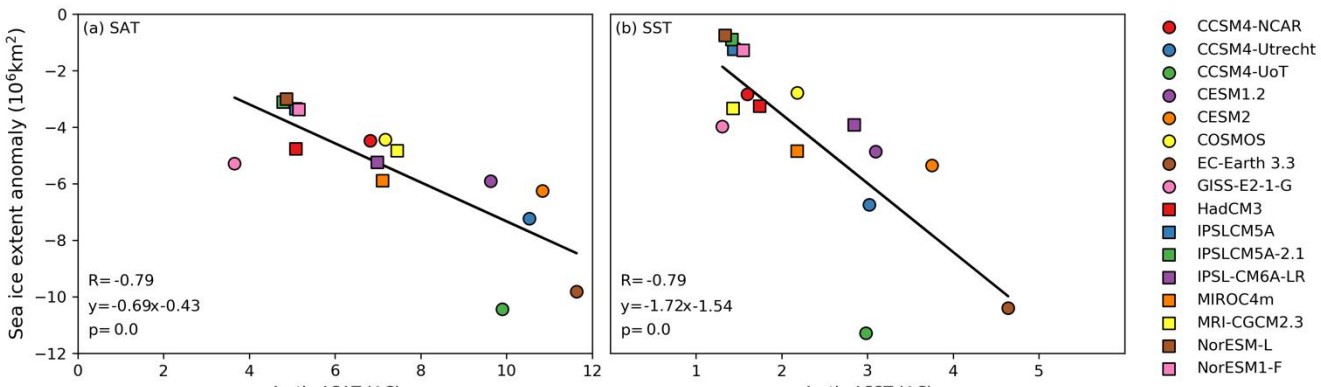

**Figure 6: Correlations between annual mean SIE anomalies and (a) Arctic SAT anomalies and (b) Arctic SST anomalies.**

## 5 Data-model comparison surface air temperatures

### 5.1 Results


To evaluate the ability of the PlioMIP2 ensemble to simulate Arctic warming, we perform a data-model comparison with the available SAT reconstructions for the mPWP. The data-model comparison hints at a substantial mismatch between models and temperature reconstructions. Mean absolute deviations (MAD) range from 5.0 to 11.2 °C (Table S3), with a MAD of 7.3 °C for the MMM. The median bias ranges from -2.0 to -13.1 °C, with a median bias of -8.2 °C for the MMM (Table S3). The

PlioMIP2 MMM shows slightly improved agreement with the SAT reconstructions compared to the PlioMIP1 MMM (MAD = 7.8 °C, median bias = -8.7 °C). Figure 7 depicts the deviation from reconstructions for the MMM. Underestimations range from -17 to -2.5 °C, while at two sites (80° N, 85° W and 79.85° N, 99.24° W) the MMM overestimates the reconstructed temperatures (by 2.7 and 1.2 °C, respectively).





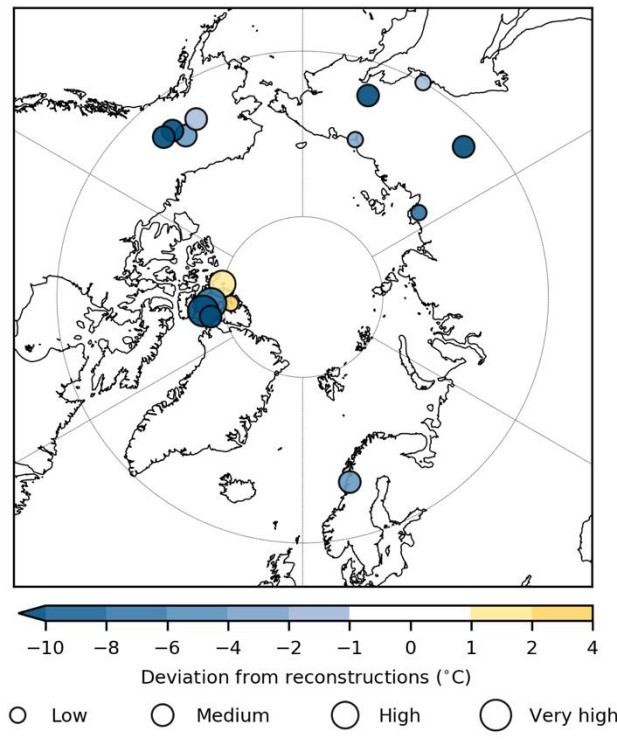

**Figure 7: Point-to-point comparison of MMM and reconstructed SAT. The size of SAT reconstructions is scaled by qualitatively assessed confidence levels (Salzmann et al., 2013). Data markers for reconstructions in close proximity of each other have been slightly shifted for improved visibility.**

The deviation from reconstructions for each model and the PlioMIP2 and PlioMIP1 MMMs is represented by the box-whisker plots in Fig. 8. A consistent underestimation of the temperature estimates from SAT reconstructions is present in the PlioMIP2 ensemble. CESM2 simulates the smallest deviations from reconstructions in the ensemble, with a MAD of 5.0 °C and a median bias of -2 °C. The five models with the highest Arctic SAT anomalies (CCSM4-Utrecht, CCSM4-UoT, CESM1.2, CESM2, and EC-Earth 3.3) simulate the five lowest median biases in the ensemble, indicating that the upper end of the range of simulated Arctic SAT anomalies in the PlioMIP2 tends to match proxy dataset in its current form better.





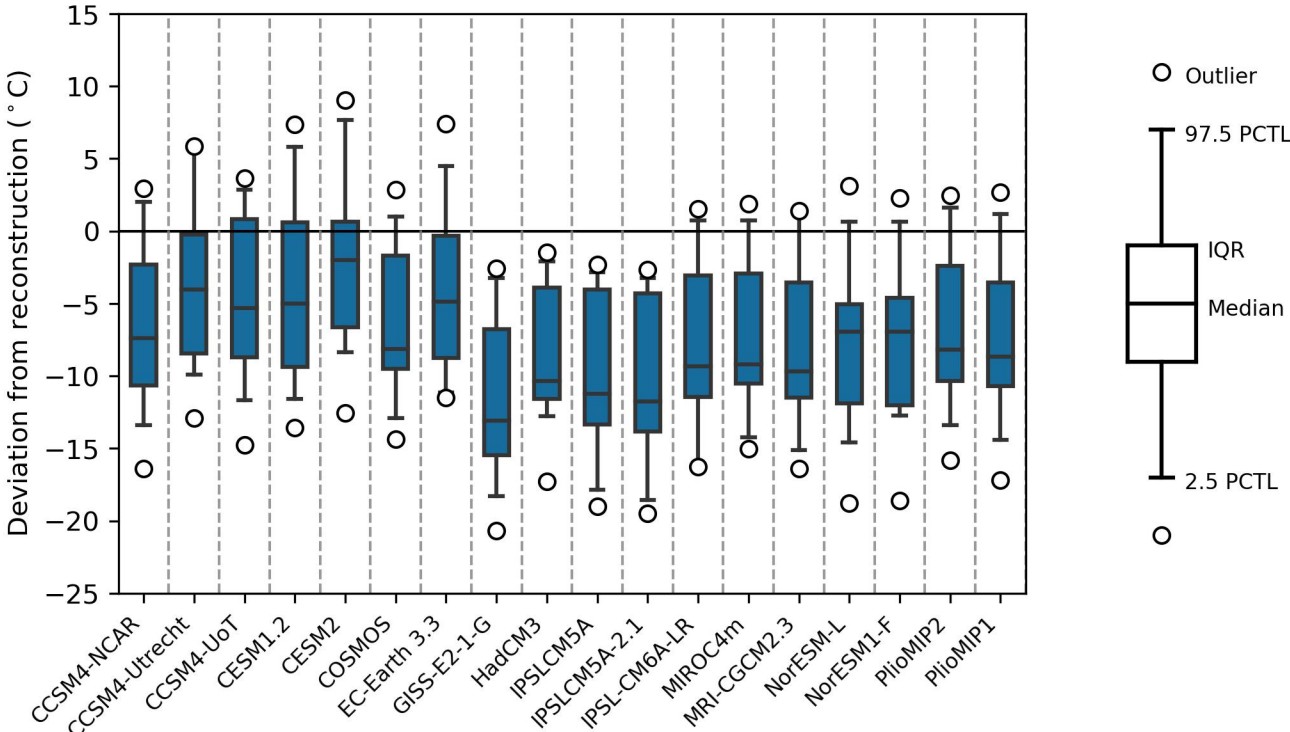

**Figure 8: Box-whisker plots depicting the distribution of biases (models minus reconstruction) with biases over (under) 0 representing locations where models overestimated (underestimated) reconstructed temperatures. Boxes depict the interquartile ranges (IQR) of the distribution, whiskers extend to the 2.5th and 97.5th percentiles, the median is displayed by a horizontal line in the boxes, and outliers (outside of the 97.5th percentile) by open circles outside of the whiskers. Given the sample size of 15 reconstructions, the two outer values are depicted as outliers using these definitions.**

## 5.2 Uncertainties

Some of the data-model discord may be caused by uncertainties in the temperature estimates (Table S1; Salzmann et al., 2013). To investigate how these uncertainties may have affected the outcomes of the data-model comparison, we calculate, similar to work by Salzmann et al. (2013), a maximum uncertainty range using estimates of temporal and bioclimatic range uncertainties.

Figure 9 depicts the locations for which at least one model in the ensemble simulates a temperature within the maximum available uncertainty range of a reconstruction. For six out of the twelve reconstructions that included an uncertainty estimate, the models in the PlioMIP2 ensemble simulate temperatures that are within the uncertainty range (Fig. 9). Additionally, both over- and underestimations are present for the Magadan District reconstruction for which no uncertainty estimate is available (60° N, 150.65° E, Table S1), implying that the reconstruction falls within the range of simulated



temperatures in the PlioMIP2 ensemble. For the remaining six reconstructions, including several which are assessed high or very high confidence (Figure 9), no model simulates temperatures within the uncertainty range.

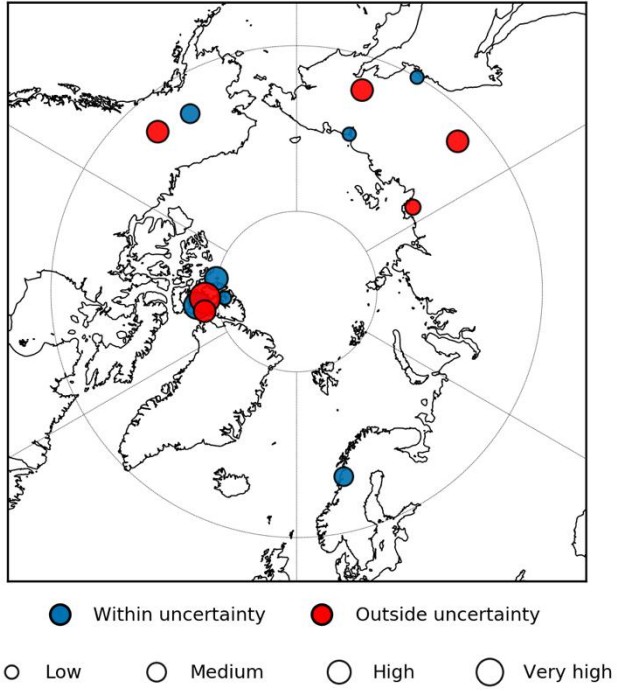

**Figure 9: Blue circles highlight where at least one model in the ensemble simulates a temperature that falls within the uncertainty range. The size of SAT reconstructions is scaled by qualitatively assessed confidence levels (Salzmann et al., 2013). Data markers for reconstructions in close proximity of each other have been slightly shifted for improved visibility.**

Ultimately, when considering the full uncertainty ranges of the reconstructions, it becomes evident that solely reducing potential errors in SAT estimates would not fully resolve the data-model discord for several locations in the Arctic. It is thus likely that other sources of error contribute to the data-model discord, such as uncertainties in model physics (e.g. Feng et al., 2019; Howell et al., 2016b; Lunt et al., 2020; Sagoo and Storelvmo, 2017; Unger and Yue, 2014) and boundary conditions (e.g. Brierley and Fedorov, 2016; Feng et al., 2017, 2017; Hill, 2015; Howell et al., 2016b; Otto-Bliesner et al., 2017; Prescott et al., 2014; Robinson et al., 2011; Salzmann et al., 2013). The focus on the KM5c time slice has helped resolve some of the data-model discord that was present in the North Atlantic for SST (Haywood et al., 2020), and similar work for SAT reconstructions may thus be beneficial. However, this may not always be possible given the lack of precise dating and chronologies available. It is at this moment unclear whether the underestimation of Arctic SAT is specific to the mid-Pliocene, through uncertainties in reconstructions or boundary conditions, or an indicator of common errors in model physics.



# 6 Evaluation of sea ice

Figure 10a depicts the number of models per grid box that simulate perennial sea ice. Six models simulate the inferred perennial sea ice (mean sea ice concentration ≥ 0.15 in each month) at Lomonosov Ridge (87.5° N, 138.3° W; Darby, 2008),

while the remaining ten simulate sea ice-free conditions in at least one month per year at this site. The majority of the models simulate a maximum SIE that extends, or nearly extends, into the Fram Strait and Iceland Sea (Figure 10b) in at least one month (in winter) per year (Fig. 10b), consistent with proxy evidence (Clotten et al., 2018; Knies et al., 2014).

Three models: HadCM3, IPSLCM5A, MRI-CGCM2.3, simulate both the perennial sea ice at Lomonosov Ridge and a sea

ice minimum that extends into the Fram Strait and Iceland Sea. MRI-CGCM2.3, however, simulates perennial sea ice in the Fram Strait, which does not correspond to seasonally sea ice-free conditions at this site that are inferred from the reconstruction (Knies et al., 2014). Ultimately, only HadCM3 and IPSLCM5A simulate sea ice that is consistent with all three available sea ice reconstructions. Interestingly, in an analysis of simulated sea ice in the PlioMIP1 ensemble, the HadCM3 simulation was also found to be consistent with proxy evidence, and it was the only model fulfilling this criterion

(Howell et al., 2016a). The HadCM3 and IPSLCM5A simulations, however, do not show improved agreement with regard to SAT reconstructions than the PlioMIP2 MMM (Fig. 8), indicating that matching the sea ice reconstructions does not necessarily imply a better match with the SAT reconstructions.

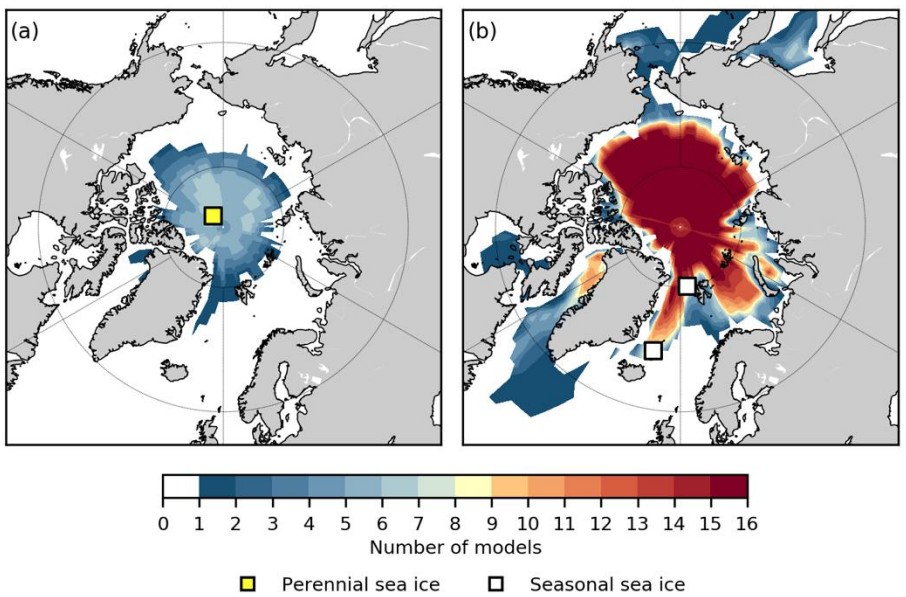

**Figure 10: Number of models simulating (a) annual mean perennial sea ice (sea ice concentration of ≥ 0.15) at any given location in**
**the Arctic in the mPWP simulations and (b) monthly mean sea ice in any month of the year.**



The uncertainties in both the SAT and SIE reconstructions are large, and it may not be possible to match both datasets in their current forms. This would require increased Arctic annual terrestrial warming compared to the mean model (Sect. 5.1) as well as perennial sea in the summer and a large SIE in winter (extending at least into the Iceland Sea). Moreover, McClymont et al. (2020) found that the warmest model values in the PlioMIP2 ensemble tend to align best with North

Atlantic SST reconstructions, further indicating that strong Arctic warming is required for data-model agreement. If there was no perennial sea ice in the mPWP like most models in the PlioMIP2 ensemble, the different proxy records may be more compatible, but this would be in disagreement with findings from (Darby, 2008). The CCSM4-Utrecht model, which simulated a relatively high Arctic SAT anomaly (10.5 °C; Figure 1a) and low median bias (-4 °C) in the point-to-point SAT data-model comparison compared to the rest of the ensemble, simulates a maximum winter SIE that extends both into the

Fram Strait and Iceland Sea. This highlights that models with higher Arctic SAT anomalies and better SAT data-model agreement can still match both seasonal sea ice proxies. Ultimately, more reconstructions of sea ice are needed for a more robust evaluation of mPWP sea ice and Arctic warming in general.

## 7 Comparison to future climates

Research into the mPWP is often motivated by a desire to understand future climate change (Burke et al., 2018; Haywood et
al., 2016; Tierney et al., 2019). Here, we analyze how the mPWP may teach us about future Arctic warming by comparing some climatic features of the mPWP simulations to simulations of future climate. The climatic features include Arctic amplification, and two features for which there is some proxy evidence available that may also aid in model evaluation: the AMOC, and the northern modes of variability.

### 7.1 Arctic amplification

A linear relationship between global and Arctic temperature anomalies is present in the PlioMIP2 ensemble (R=0.93, Fig. 11a). This is consistent with findings from multi-model analyses of other climates (Bracegirdle and Stephenson, 2013; Harrison et al., 2015; Izumi et al., 2013; Masson-Delmotte et al., 2006; Miller et al., 2010; Schmidt et al., 2014; Winton, 2008) and indicates that global temperature anomalies are a good index for Arctic SAT anomalies in mPWP simulations.

For four ensembles of future climate simulations, from the previous phase of the Coupled Model Intercomparison Project (CMIP), CMIP5, data for MMM Arctic (defined there as 67.5–90° N) temperature anomalies are available (Masson-Delmotte et al., 2013; Table S4). The PlioMIP2 MMM shows global warming that falls between the RCP6.0 and RCP8.5 MMMs in terms of magnitude (Fig. 11b). Even though PlioMIP underestimates mPWP SAT reconstructions (Sect. 5.1), the simulations do simulate stronger Arctic temperature anomalies per degree of global warming compared to future climate

ensembles (Fig. 11b). The future climate ensemble MMMs simulate Arctic (67.5–90° N) amplification ratios that range from 2.2 to 2.4, while PlioMIP2 and PlioMIP1 simulate mean ratios of 2.8 and 2.7, respectively (Table S4).



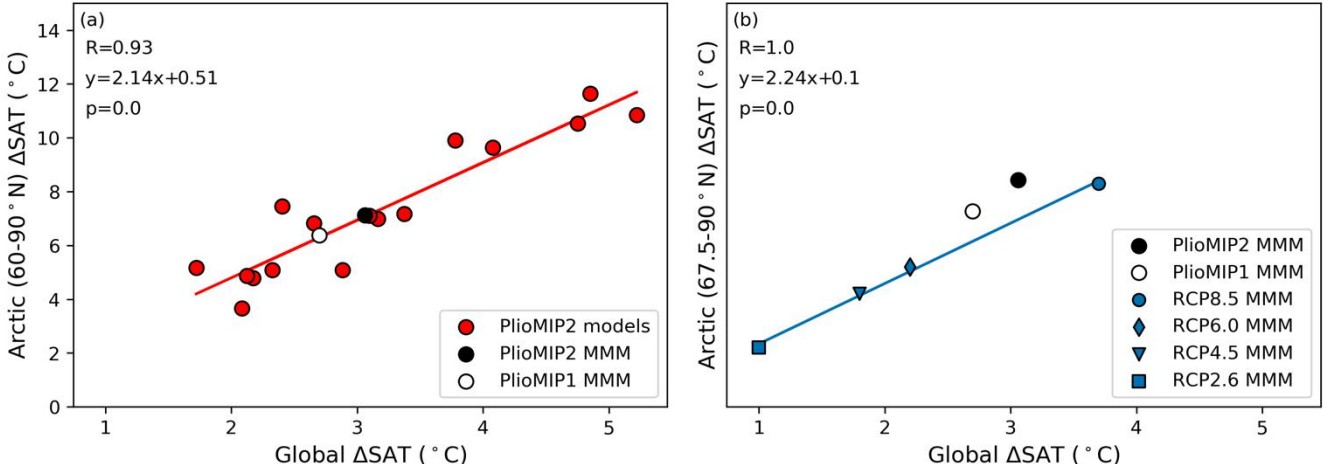

**Figure 11: (a) The relationship between global and Arctic (60–90° N) temperature anomalies in the PlioMIP2 ensemble. The red trendline is constructed based on this relationship for the individual models. (b) The relationship between global and Arctic (here 67.5–90° N due to limited data availability of the future climate simulations) for the MMMs of the two PlioMIP and the four CMIP5 future climate ensembles. The blue trendline highlights this relationship for the RCP MMMs.**

The increased Arctic warming per degree of global warming indicates that apart from warming through changes in atmospheric $CO_2$ concentrations, which is the dominant mechanism for warming in both ensembles, different or additional mechanisms underly the simulated mPWP Arctic warming compared to the future climate simulations. The difference between the PlioMIP2 and future climate ensembles may be explained by slow responses to changes in forcings that fully manifest in equilibrium climate simulations, such as the response to reduced ice sheets, but not in transient, near-future, climate simulations. Additional Arctic warming in the mPWP simulations may arise due to the changes in Arctic ocean gateways (Brierley and Fedorov, 2016; Feng et al., 2017; Haywood et al., 2016; Otto-Bliesner et al., 2017), other changes in orography, and vegetation in the boundary conditions (Hill, 2015; Lunt et al., 2012b). These findings highlight the caution that has to be taken when using palaeoclimate simulations as analogues for future climate change.

### 7.2 Atlantic meridional overturning circulation

The AMOC, a major oceanic current transporting heat into the Arctic (Mahajan et al., 2011), is inferred to have been significantly stronger in the mPWP compared to pre-industrial based on proxy evidence (Dowsett et al., 2009; Frank et al., 2002; Frenz et al., 2006; McKay et al., 2012; Ravelo and Andreasen, 2000; Raymo et al., 1996). An analysis of AMOC changes in PlioMIP2 simulations shows that, indeed, the maximum AMOC strength increases: by 4 to 53% (Fig. 12; Table S2: Li et al., in prep.). The closure of the Arctic Ocean gateways, in particular the Bering Strait, likely contributed to the increase in AMOC strength (Brierley and Fedorov, 2016; Feng et al., 2017; Haywood et al., 2016; Otto-Bliesner et al., 2017).

Strengthening of the AMOC contrasts projections of future changes by CMIP5 models which predict a weakening of the AMOC over the 21[st] century, with best estimates ranging from 11 to 34% depending on the chosen future emission scenario





(Collins et al., 2013). These opposing responses may help explain some of the additional Arctic warming that is observed in the PlioMIP2 ensemble (Fig. 11b). This is consistent with the additional 0.4 °C increase in SST warming in the Arctic (Figure 1c) and the better data-model agreement in the North Atlantic that is observed for the PlioMIP2 MMM (Dowsett et al., 2019; Haywood et al., 2020; McClymont et al., 2020) compared to the PlioMIP1 MMM (Fig. 1c), which did not show 405 any substantial changes in AMOC strength compared to pre-industrial (Zhang et al., 2013).

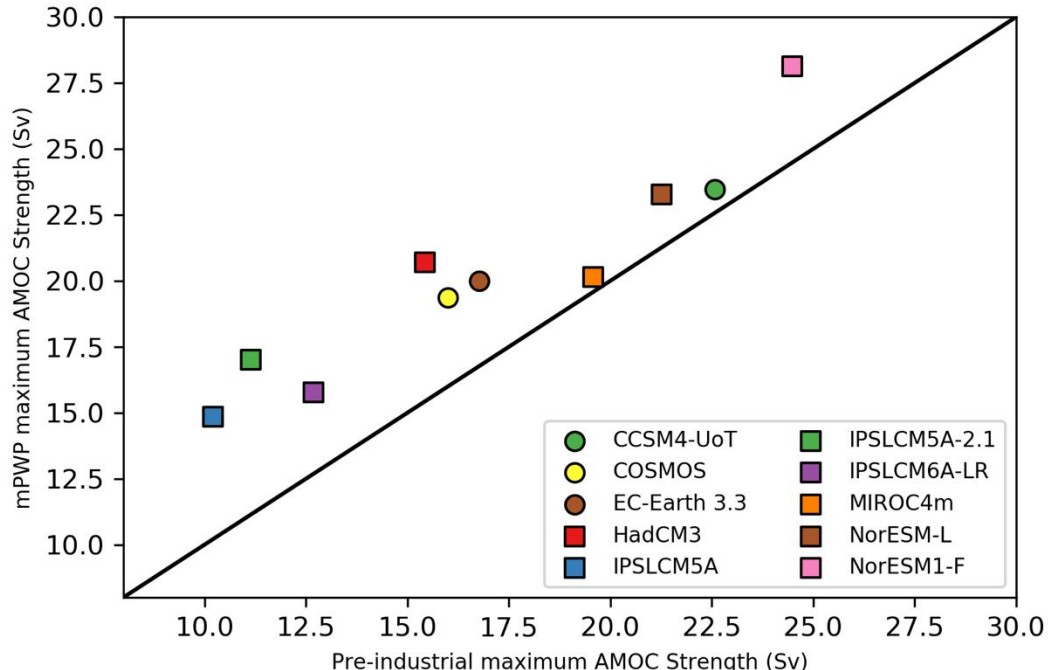

Figure 12: Maximum pre-industrial and mPWP AMOC strength (Sv). The black line indicates equal pre-industrial and mPWP maximum AMOC strength.

**7.3 Northern modes of variability**

Changes in atmospheric circulation in response to Arctic warming are key both to the understanding of Arctic climate variability and because they govern the remote consequences of Arctic amplification (Smith et al., 2019). Reconstructions of mid-Pliocene NAO based on fossilized tree rings indicate the presence of a strong NAO in the Atlantic sector of the Canadian Arctic (Hill et al., 2011). The NAO is the regional expression of the Northern Annular Mode (NAM), the dominant mode of sea level pressure variability in the Northern Hemisphere (Thompson and Wallace, 2000). Here, we analyze the 415 changes in the boreal winter NAM and NAO to identify how the changes in boundary conditions in the mPWP may have affected northern hemispheric climate variability.

We calculate the NAO as the first principal component axis (PCA) of boreal winter (DJF) mean sea level pressure (SLP) in the North Atlantic (20–90° N, 90° W–40° E; Hurrell and Deser, 2010). We calculate the NAM as the first PCA of the DJF





mean SLP field poleward of 20° N (Hurrell and Deser, 2010). The relative change in the strength of the NAO and NAM is
       calculated by taking the ratio of the standard deviation of the mPWP simulation and the pre-industrial simulation.

       The PlioMIP2 ensemble does not simulate a consistent change in the strength of the NAO as half of the ensemble simulates
       increases in the strength of the NAO, while the other half simulates decreases (Fig. 13a). Moreover, the simulated changes in
the NAO strength are mostly limited in size. The ensemble may, however, hint at a slight shift towards a stronger NAM,
       with 11 models simulating increases in the standard deviation of its PCA (Fig. 13b).

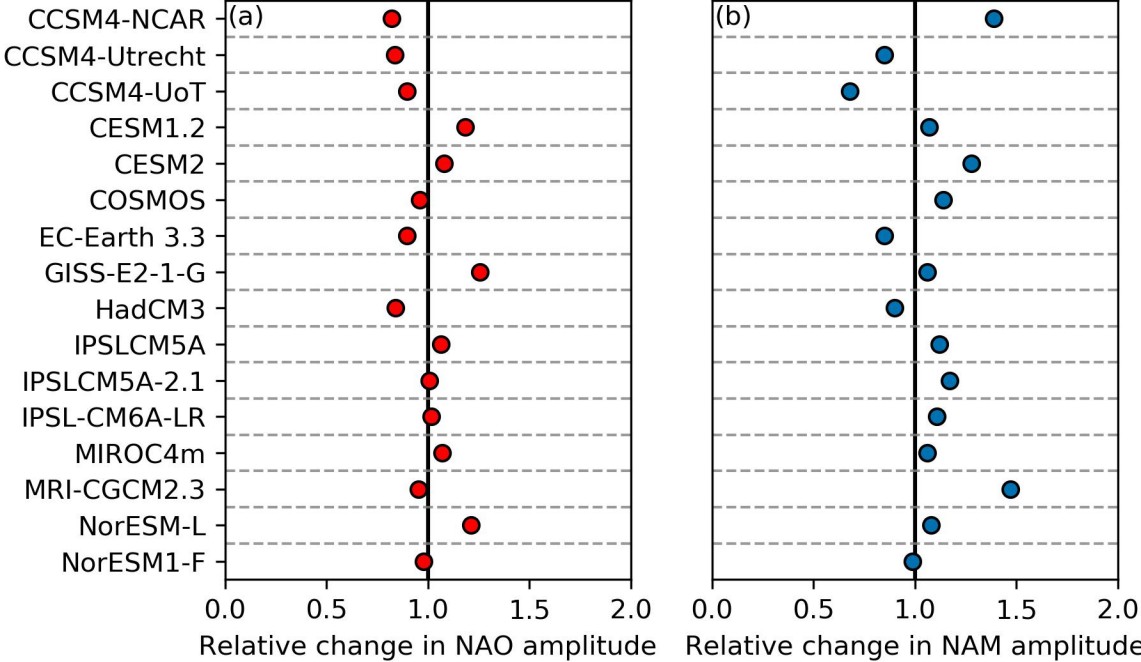

**Figure 13: The ratio of change in the amplitude of the (a) NAO and (b) NAM indices between the mPWP and pre-industrial
simulations.**

The results are compatible with the sparse proxy evidence from the mPWP, which indicates an active NAO strength (Hill et
       al., 2011). Projections of future climate change under the RCP4.5 scenario suggest an increase in NAO and NAM strength
       compared to pre-industrial, although the signal-to-noise ratio is limited (Christensen et al., 2013). While we have no
       available data for the RCP6.0 and RCP8.5 scenarios that are more similar to PlioMIP2 in terms of large-scale temperature
       response (Fig. 11b), we assume a similar response would occur in these future climates. The RCP4.5 projections are similar
to the strengthening of the NAM simulated by most of the PlioMIP2 ensemble (Fig. 13b), but not for the NAO.

       Analysis of the impact of the PlioMIP2 experiment on the average winter surface pressure gradients may help understand
       this different behaviour. We calculate a new NAM and NAO index based on the mean DJF SLP values, following the
       definitions in Christensen et al. (2013). The individual models in PlioMIP2 simulate a consistent increased NAM index (by



an average of 0.28 hPa, Fig. 14a), and an increased NAO index (by an average of +0.16 hPa, Fig. 14b). Both of these

changes are in a similar direction to that projected under the RCP4.5 scenario, although the magnitude of change is larger in

the future projections (Christensen et al., 2013). This would indicate that the changes in mean states of the underlying storm

tracks are broadly similar between PlioMIP2 and RCP4.5. Therefore, further detailed analysis is required to reconcile the

increased NAO amplitude in the RCP4.5 projections with the PlioMIP2 results (Fig. 13a), with potential explanations

involving changes in the spatial patterns of SLP variations and transient versus equilibrium effects.

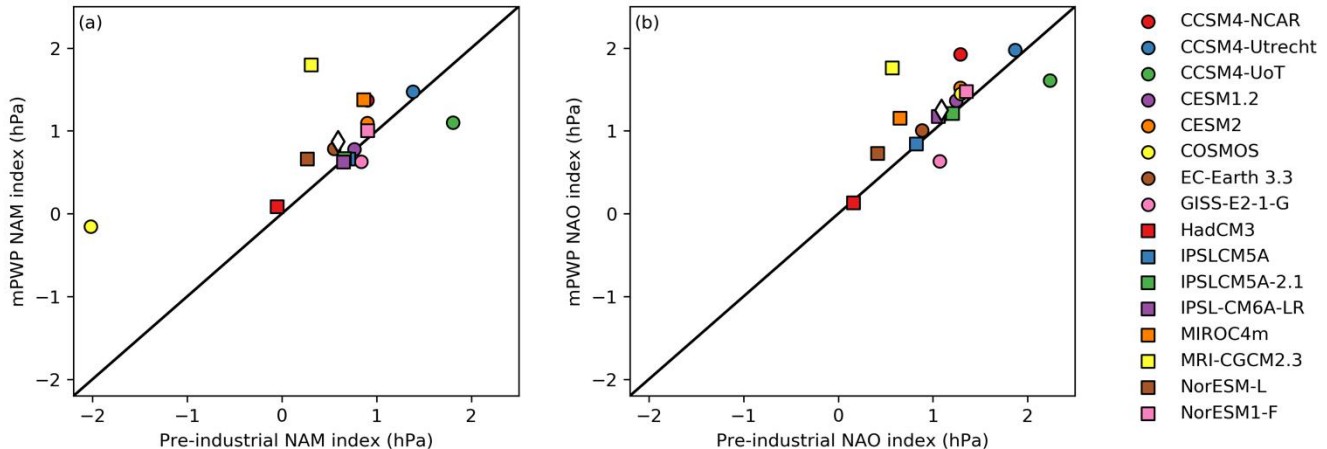

**Figure 14: Mean values of the boreal winter (a) NAM and (b) NAO indices following the definitions in Christensen et al. (2013). The NAM calculated as the difference in zonal mean SLP at 35°N minus 65° N (Li and Wang, 2003) and the NAO as the difference of regional averages (90° W to 60° E, 20° N to 55° N) minus (90° W to 60° E, 55° N to 90° N) (Stephenson et al., 2006). The black line indicates equal pre-industrial and mPWP anomalies. The average indices for the 16 models is denoted by the white diamond.**


## 8 Conclusions

The PlioMIP2 ensemble simulates substantial Arctic warming and 11 models simulate summer sea ice-free conditions.

Comparisons to reconstructions show, however, that the ensemble tends to underestimate the available reconstructions of

SAT in the Arctic. We find that, while some of the SAT data-model discord may be resolved by reducing uncertainties in

proxies, additional improvements are likely to be found in enhanced boundary conditions or model physics. Furthermore,

there is some agreement with reconstructions of sea ice in the ensemble, especially for seasonal sea ice. The limited

availability of proxy evidence and the uncertainties associated with them severely constrain the compatibility of the different

proxy datasets and our ability to evaluate the Arctic warming in PlioMIP2. Increased proxy evidence of different climatic

variables, and additional sensitivity experiments, among others, are needed for a more robust evaluation of Arctic warming

in the mPWP. Lastly, when considering the mPWP as a potential analogue for future climate change, firstly the incomplete

manifestation of slow responses in transient simulations, and secondly the observed differences in Arctic climate features

between the ensembles, including the magnitude of Arctic amplification, and the changes in AMOC strength and northern

modes, which show some agreement with proxy evidence, should be taken into account.



*Supplement.*

*Author contributions.* Qiong Z. and Wesley de Nooijer designed the work, Wesley de Nooijer did the analyses and wrote the manuscript under supervision from Qiong Z., Q. L. and Qiang Z. performed the simulations with EC-Earth3. X. L. and Z.Z. provided input on AMOC analysis. C. B. provided input on NAO interpretation. H. D. provided the input on reconstructions. All the other co-authors provided the PlioMIP2 model data and commented on the manuscript.

*Competing interests.* The authors declare that they have no conflict of interest.

*Acknowledgements.* This work is supported by the Swedish Research Council (VR) funded projects 2013-06476 and 2017-04232. The EC-Earth3 simulations are performed on the Swedish National Infrastructure for Computing (SNIC) at the National Supercomputer Centre (NSC). COSMOS PlioMIP2 simulations have been conducted at the Computing and Data Center of the Alfred-Wegener-Institut– Helmholtz-Zentrum für Polar und Meeresforschung on a NEC SX-ACE high

performance vector computer. G. L. and C. S. acknowledge funding via the Alfred Wegener Institute's research programme PACES2. C. S. acknowledges funding by the Helmholtz Climate Initiative REKLIM. C.C. and G.R. thank ANR HADOC ANR-17-CE31-0010. Authors were granted access to the HPC resources of TGCC under the allocations 2016-A0030107732, 2017-R0040110492 and 2018-R0040110492 (gencmip6) and 2019-A0050102212 (gen2212) provided by GENCI. The IPSL-CM6 team of the IPSL Climate Modelling Centre (https://cmc.ipsl.fr) is acknowledged for having developed, tested,

evaluated, tuned the IPSL climate model, as well as performed and published the CMIP6 experiments. W.-L.C. and A.A.-O. acknowledge funding from JSPS KAKENHI grant 17H06104 and MEXT KAKENHI grant 17H06323, and JAMSTEC for use of the Earth Simulator supercomputer. The PRISM4 reconstruction and boundary conditions used in PlioMIP2 were funded by the U.S. Geological Survey Climate and Land Use Change Research and Development Program. Any use of trade, firm, or product names is for descriptive purposes only and does not imply endorsement by the U.S. Government.

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
