# Peer review of "Evaluation of Arctic warming in mid-Pliocene climate simulations"

_Climate of the Past, 2020_

## Referee Comment (RC1) · Anonymous Referee #1 · 19 May 2020

Review of "Evaluation of Arctic warming in mid-Pliocene climate simulations" by de Nooijer et al.

The authors provide a good and well written summary of several aspects of the results of the latest round of Pliocene simulations. These simulations and their comparison with available geological records are important since this period is one of the few that provides an estimate of climatic changes that are to first order driven by changes in greenhouse-gas concentrations.

In the manuscript there are several aspects that should be looked at more closely and some that should be discussed more clearly. Below I will detail my concerns.

[Figure]

Main concerns:

Impact and/or importance of values orbital parameters:
Lines 118- 130: The authors mention that for the PlioMIP2 simulations a specific time-slice was chosen in order to have values of the orbital parameters that are similar to today. The shorter orbital cycles are 20 and 40 kiloyears, meaning that an uncertainty in the estimate age of a mPWP temperature reconstruction of 10 kiloyear could already imply quite different values of the orbital parameters. I'm not an expert on that topic, but it seems to me the age constraints that are needed to make a firm statement about the orbital parameters that accompany the climate reconstructions that are used in this work, are very difficult to obtain.

The authors also mention various experiments that have been done for the Pliocene to investigate the impact of different orbital parameters. Could those results be combined with the model-data comparison provided in this paper for a more extensive discussion on the topic?

Uncertainty of proxy-based climate reconstructions:
Lines 301-304: Please shortly reiterate how this maximum uncertainty range is estimated as it is quite important for the discussion that follows. Does it for instance include any discussion on the interpretation of the climate reconstructions? Any seasonal biases? From reading the referenced literature it appears that changes in for instance the growing season are considered important drivers of the temperature reconstructions, but I don't see a discussion on this topic in the current paper. How strong is the evidence that the reconstructed temperatures reflect changes in the annual mean rather than a value that is biased towards certain seasons?
To investigate the importance of this issue, many studies resort to comparing the

temperature reconstructions with both simulated annual mean and simulated summer temperatures, has that been considered?

Robust changes in NAO and/or NAM?
Line 435: Please be more clear about whether the RCP4.5 simulations show robust changes in NAO and/or NAM. Do you have grounds to conclude that this is the case for the PlioMIP2 simulations?
Similarly on lines 436-445: Are the changes in NAM and NAO significant? So it depends on the metrics that is used to calculate these modes of variability whether or not the changes are significantly? What does that mean? And while the temperature changes in the RCP4.5 simulations are smaller, the NAO/NAM changes are larger? Please clarify.

Minor comments:

Lines 124-130: So how many models did actually close the Bering Strait? From figure 2 it can be concluded that not all did, but you mention that this change in experimental design improved the model-data fit so it is important to state this clearly.

Lines 270-271: How is this conclusion reached? Why is it not important to correctly simulate SAT anomalies for the SIE anomalies?

Lines 334-357: Of course the authors realize that having only three data points in the whole Arctic Ocean doesn't make for a particularly strong model-data comparison, but we have to work with what we have. Nonetheless, the text should clearly reflect this. The site in the Iceland Sea appears to be very close to the boundary between the regions that are never covered by sea ice and those that are covered at least one month a year. One cannot expect a course resolution climate model to

put this boundary at the exact right location and thus no strong conclusions can be attached to a model-data comparison at such a site.

Lines 382-390: the authors should more clearly state what the differences are between the paleo and future simulations. Both are forced with greenhouse-gas concentration changes, but the paleo runs are further forced by changes in the ice-sheets, vegetation, gate-ways? As for the changes in ice-sheets, vegetation and also the AMOC, one could argue that these simulation give a true long-term equilibrium response to greenhouse-gas changes. This is not the case for the impact of changing the Arctic gate-ways. Is there a way to quantify the impact of the latter as to make the comparison with future simulations more meaningful?

Lines 391-398: There are a number of studies discussing simulations of the impact of closing the Bering Strait on the AMOC strength, do they also show a moderate to strong increase in AMOC strength?

Line 430: what is meant with an 'active NAO strength'? It appears that the models do not provide robust support of a change in NAO amplitude.

Line 433: Why are RCP6.0 and RCP8.5 simulations not used in the comparison if those provide a better comparison in terms of temperature changes?

Line 455: What would such improvements in boundary conditions be? Don't the authors think that all changes in boundary conditions that are likely to have a significant effect are already included?

Figure 11: limited data availability? The data between 60N and 67.5N is missing?

Technical comments:

Line 110: For me forcings are not part of model physics. Please clarify.

Line 145: missing space

Line 451: "11 out of 16", just for clarity.

Table 1: It would be good to add to this table if the model is also used in CMIP5, CMIP6 or neither of those.

Caption figure 3: shouldn't that be "compared to the annual mean in a given month"?

Figure 6: what does the 'p' stand for?

Figure 11: What is shown for the RCP simulations, an average over year xx to yy?

---

## Referee Comment (RC2) · Anonymous Referee #2 · 7 Jun 2020

Review of manuscript "Evaluation of Arctic warming in mid-Pliocene climate simulations " - de Nooijer et al.

In the present manuscript, de Nooijert et al. present an analysis of Arctic climate as simulated by the coupled models ensemble from the PLIOMIP2 initiative. PLIOMIP2 focuses on the specific KM5c interval, peak of the mPWP. Notable improvements have also being done for the boundary conditions (e.g. closed Arctic gateways during this period). Models generally simulate an Arctic amplification larger than 2.5, increase in SAT and SST. Comparison with the few existing proxies suggest that only few models of the ensemble are able to fit the warm climatic conditions of the particular KM5c interval. However, the lack of proxies prevent a more detailed comparison. An attempt is made to compare those new results to projections. Conclusion of the authors is that

using the simulated mPWP KM5c is not yet informative for the future, given the current state of models and limitations of the design of the experiments and lack of proxies to validate the paleo-simulations.

In general, what this phase 2 of the PLIOMIP initiative shows is that boundary conditions improvements and focus on a specific interval of the mPWP generally increase the agreement with the few existing proxies. However, the paper remains rather very elusive and not detailed too much about the causes of the simulated anomalies. In addition, there is a distinct dichotomy within the models with only few models increasing the MMM. An aspect that is really unclear throughout this manuscript is the impact of the models that do not used closed gateway in their simulations and how much this impact on the interpretation of the entire metrics presented here. In addition to closed gateways, individual model resolution might also have an impact on the representation of those gateways and this is not discuss here. The attempt made to compare with CMIP5 projections is to my opinion unsuccessful given the striking difference in gateways between the modern geography and that of the mPWP. In addition, the authors attempt to compare the mode of variability which is a non-sens here since the paleoclimatic simulations are equilibrium simulations while projections are transient short-term simulations. Authors warn about the lack of "slow-feedbacks" in the projections, but the contrary is also true, the short-term variability present some limitation in the paleoclimate runs. I do not advise to remove it. However, some improvements are needed to strengthen those parts and to make them meaningful in a way or in the other.

The manuscript is written quite well (though in some places that I have indicated in my comments below, some improvements in the writing is needed to clarify). My impression is that this paper remains superficial and does not provide a real analysis of the Arctic warming. There is no real analysis of the causes/consequences of this warming (i.e. albedo, seasonal cycle in temperature, snow cover, westerlies etc.)... Even if the number of proxies is limited, the authors could deepen their analysis to compare the different models together to provide partial answers to some of the questions posed

by the authors themselves within the different sub-section of the manuscript. They should also explore the dichotomy amongst the models visible in almost all the figures of this manuscript and the impact this dichotomy has on the MMM and thus the overall interpretation of the MMM. I therefore recommend moderate revisions.

Comments:

Line 68: I would remove "future" and just write "as warming in the Arctic directly affects...". This is because this is always true, not only for future. Or perhaps just reformulate in "as it is shown that projected Arctic warming affects...".

Line 84: Would it be worth mentioning that the interest of the KM5c interval is because orbitals are similar to present? I think this is important and relevant to the comparison with projections.

Line 141: correct "model resultsaere calculated" in "model results are calculated"

Line 196-203: I find interesting to note that most of the models simulate air and sea temperature values below the mean and that only a couple of models exhibit values much higher than the mean. It could also be worth mentioning this somewhere (though it is not a paper about individual model performances) because it also impacts on the interpretation that one does about the ensemble mean.

Line 209: but did not you write that also the Bering Strait is closed in some of the models? We don't see a particularly large anomaly around this area.

Line 212: and thus? What causes such an increase in the Baffin Bay? The lack of sea ice due to no arctic waters flowing through the CA? If yes, it would be good to mention.

Line 196 - 215: How does the discrepancy in land sea mask, especially in the Bering Strait, affect the interpretation of the MMM in Figure 2? I would find very informative to indicate which models closed the Bering Strait and or the Canadian archipelago in Table 1. It seems from Figure 2b that only a few models keep the Bering Strait open. Are the models with open Bering Strait the ones with highest SST and SAT values (e.

g. In Fig.1)?

Lines 272-289: How much is the MMM-proxy comparison valid in the Canadian archipelago? I mean, in Figure 7 the proxies there are very closed to each others (while already slightly shifted for better understanding) and, how many grid points are there in in the simulations this area? Is the comparison here valid? Or not resolution-dependent? Same for Alaska?

Figure 8: Since the beginning, there are two distinct groups amongst the models and the MMM is shifted to higher value because of 7 models. This discrepancy between the two groups is very neat. Thus I really wonder what are the causes of such dichotomy and what is the impact on the interpretation of the MMM in the paper in general?

Lines 320-321: but also models should also all use the same boundary conditions. Because if some fo the models do not close some fo the straits, or if they have no sufficient resolution to capture the width of some passages etc. . .how can we interpret the misfit between data and models correctly? I mean, as it is now, it is impossible to determine wether or not in some models the different boundary conditions or different physics affect the misfit and in which proportion. I know it is very difficult to modify the land-sea mask in coupled models and in some cases it will also require more computational resources to increase spatial resolution enough to capture the different gateways properly. However, at some points, we will need to do it to further advance those types of data-model exercises.

Figure 10: yellow and white squares are reconstructions from proxies? I guess yes. . . but this is not mentioned in the caption.

Figure 11: is the vertical Y scale in frame b) the same as in frame a)? In any case, please add the ticks for dSAT values on the graph for projections.

Lines 377- 381: When reading those lines, it seems that only CO2 forcing matters here. But in many of your models, some gateways are closed, and as you cite Otto-Bliesner

et al. (2017), this matters. . . Thus I disagree with the formulation of those sentences. Please also discuss the difference in Arctic geography and how this impact ton the comparison with the projections.

Lines 396-400: Given the different boundary conditions, I find very difficult to make a direct comparison here. In most of PLIOMIP2 models, the Arctic gateways are closed and this generates a strengthening of the AMOC. While under modern geography, the Arctic gateways are open and a weakling of the AMOC is projected. You cannot compare those two situations here directly. In general, this short paragraph is not very clear. If you state more clearly at the beginning and in Table 1 that not all models prescribed closed gateway, this would definitely improve the reader understanding of the paper.

Line 397: "This is consistent" To what does "this" refer to?

Subsection 7.3: To my opinion, it is very difficult to compare transient short-term projections variability with equilibrium climate variability of a few centuries (as just say line 440). Thus I find not very much straight forward and informative the conclusions from this comparison here.

Lines 427-429: This sentence is very unclear, please reformulate.

Lines 455 - 458: You state about the discrepancies between mPWP and projections simulations: "firstly the incomplete manifestation of slow responses in transient simulations". But not only, I would say also vice-versa: "the lack of transient variability in equilibrium climate". Then you state "secondly the observed differences in Arctic climate features between the ensembles": which ensembles are you referring too here? PLIOMIP1 versus PLIOMIP2 or PLIOMIP2 versus projections? If this is the second option, then I would say the entire sentence does not make sense because of course they are different, besides equilibrium versus transient, boundary conditions also differ. . .

---

## Author Comment (AC1) · 3 Aug 2020

Thanks for the reviewer's constructive comments which lead to an improved manuscript, below is our point to point response. We also attached this author response in the supplement PDF file for you to refer.

Review comment: The authors provide a good and well written summary of several aspects of the results of the latest round of Pliocene simulations. These simulations and their comparison with available geological records are important since this period is one of the few that provides an estimate of climatic changes that are to first order driven by changes in greenhouse-gas concentrations. In the manuscript there are several aspects that should be looked at more closely and some that should be discussed

more clearly. Below I will detail my concerns.

Main concerns: Impact and/or importance of values orbital parameters: Lines 118-130: The authors mention that for the PlioMIP2 simulations a specific time-slice was chosen in order to have values of the orbital parameters that are similar to today. The shorter orbital cycles are 20 and 40 kiloyears, meaning that an uncertainty in the estimate age of a mPWP temperature reconstruction of 10 kiloyear could already imply quite different values of the orbital parameters. I'm not an expert on that topic, but it seems to me the age constraints that are needed to make a firm statement about the orbital parameters that accompany the climate reconstructions that are used in this work, are very difficult to obtain. The authors also mention various experiments that have been done for the Pliocene to investigate the impact of different orbital parameters. Could those results be combined with the model-data comparison provided in this paper for a more extensive discussion on the topic?

Reply: The reviewer is correct that the age estimates of the reconstructions are not resolved to the temporal resolution required to state that the reconstructions represent a specific set of orbital parameters, such as the similar-to-modern parameters within the KM5c time slice. In the introduction section, we mention that the focus on the KM5c time slice was useful for SST data-model comparisons, as SST estimates could be resolved to that resolution.

This resolution is not (currently) possible for SAT estimates. We mention the uncertainties with the SAT estimates in the methods, and reiterate in the conclusions that our ability to evaluate the Arctic SAT anomalies is constrained by the limited availability and uncertainties of the reconstructions. No changes were made.

However, it is an interesting suggestion to incorporate the results of other studies to see what the magnitude of the errors due to different orbital parameters could be. Feng et al. (2017) investigated the effects of changing orbital parameters, by performing sensitivity experiments that included respectively the minimum and maximum

possible insolation at 65N in July. In their conclusions they mention "Individual forcings of elevated CO2 level (by 50 ppm), high summer/annual insolation of NHL, and closed Arctic gateways may explain 1–2 °C of the terrestrial model-proxy data mismatch in the NHL." (NHL=Northern high latitudes). We added a sentence that includes these results to give an impression of the magnitude of error associated with the orbital parameters.We add at Line 174-176: "Feng et al. (2017) investigated the effects of different orbital configurations, as well as elevated atmospheric CO2 concentrations (+50ppm) and closed Arctic gateways in PlioMIP1 simulations, and found that they may change the outcomes of data-model comparisons in the northern high latitudes by 1-2 °C."

Review comment: Uncertainty of proxy-based climate reconstructions: Lines 301-304: Please shortly reiterate how this maximum uncertainty range is estimated as it is quite important for the discussion that follows. Does it for instance include any discussion on the interpretation of the climate reconstructions? Any seasonal biases? From reading the referenced literature it appears that changes in for instance the growing season are considered important drivers of the temperature reconstructions, but I don't see a discussion on this topic in the current paper. How strong is the evidence that the reconstructed temperatures reflect changes in the annual mean rather than a value that is biased towards certain seasons? To investigate the importance of this issue, many studies resort to comparing the paper temperature reconstructions with both simulated annual mean and simulated summer temperatures, has that been considered?

Reply: Changed Line 303 to reiterate how we calculate the maximum uncertainty range: "To investigate how these uncertainties may have affected the outcomes of the data-model comparison, we calculate the minimum and maximum temperature within the uncertainty, using the uncertainties for the temperature estimates as given by Feng et al. (2017)."

In the methods we state: "Reconstructed mPWP SATs are taken from Feng et al. (2017), who updated and combined an earlier compilation made by Salzmann et al.

(2013) (Table S1). Hence, the uncertainties were all indirectly derived, they were derived from compilations. It is beyond the scope of this paper to investigate these uncertainties further. For clarity, we add later in the paragraph the following sentence: "The uncertainties in the reconstructions were derived by Feng et al. (2017) and Salzmann et al. (2013) from relevant literature."

Good point about the potential bias towards seasons. As mentioned above, we will not go into this in detail but it is worth mentioning. In the following sentence: "Further uncertainties arise due to bioclimatic ranges of fossil assemblages, errors in pre-industrial temperatures from the observational record, and additional unquantifiable factors." We add "potential seasonal biases". (At the end of this paragraph we refer to Salzmann et al. (2013) for a more detailed description of the uncertainties).

While it would definitely be interesting to compare the results to summer temperatures, in the discussion we merely try to give an indication of how the magnitude of the uncertainties associated with the reconstructions may have affected the outcomes of the data-model comparison, rather than investigating the causes and validity of these uncertainties.

Review comment: Robust changes in NAO and/or NAM? Line 435: Please be more clear about whether the RCP4.5 simulations show robust changes in NAO and/or NAM. Do you have grounds to conclude that this is the case for the PlioMIP2 simulations?

Reply: Upon further inspection and thorough discussion, we decide to remove the section about the NAO/NAM. Based on comments of both Reviewer 1 and Reviewer 2. With the following reasons: - The results for both the PlioMIP2 and the RCP4.5 simulations are not very robust. There is a low signal-to-noise ratio. - The comparison of the PlioMIP2 and RCP4.5 simulations is significantly hindered by the different nature of the simulations: Equilibrium versus transient. As pointed out by reviewer 2. - The comparison is further hindered by the potential strong effect orography has on Arctic variability in the mPWP simulation. Hill et al. (2011) ascribed most of the change in

the NAM they observed in the mid-Pliocene simulation to changes in orography. Since the changes in orography in PlioMIP2 are non-analogous with future climate change we do not feel that this comparison is useful.

We therefore remove Section 7.3, and make appropriate changes in the abstract, introduction, the start of Section 7, and the conclusions to represent this.

Review comment: Similarly on lines 436-445: Are the changes in NAM and NAO significant? So it depends on the metrics that is used to calculate these modes of variability whether or not the changes are significantly? What does that mean? And while the temperature changes in the RCP4.5 simulations are smaller, the NAO/NAM changes are larger? Please clarify.

Reply: Thank you for your comments. We refer to the comments above for our response.

Minor comments:

Review comment: Lines 124-130: So how many models did actually close the Bering Strait? From figure 2 it can be concluded that not all did, but you mention that this change in experimental design improved the model-data fit so it is important to state this clearly.

Reply: Good spot. After checking, we found that all models do have a closed Bering Strait. Furthermore, a closed Bering Strait is part of both the standard and the enhanced boundary condition datasets (Haywood et al., 2016; www.clim-past.net/12/663/2016/) in PlioMIP2 and thus part of each model's simulation. Evidently, a mistake was made with the stippling. This has been updated. Stippling became redundant and hence has been removed. Description of stippling in Figure 2 caption has been removed.

Review comment: Lines 270-271: How is this conclusion reached? Why is it not important to correctly simulate SAT anomalies for the SIE anomalies?

Reply: Indeed, this conclusion cannot be reached from this data alone. It has been removed.

Review comment: Lines 334-357: Of course the authors realize that having only three data points in the whole Arctic Ocean doesn't make for a particularly strong model-data comparison, but we have to work with what we have. Nonetheless, the text should clearly reflect this. The site in the Iceland Sea appears to be very close to the boundary between the regions that are never covered by sea ice and those that are covered at least one month a year. One cannot expect a course resolution climate model to put this boundary at the exact right location and thus no strong conclusions can be attached to a model-data comparison at such a site.

Reply: Agreed, three datapoints do not make for a strong data-model comparison. At the start of the sea ice data-model comparison we added: "The limited availability of proxy evidence (three reconstructions) severely limits our ability to evaluate the simulation of mPWP sea ice in PlioMIP2 simulations. Nevertheless, a data-model comparison is still worthwhile, as the few reconstructions that are available may form an interesting out-of-sample test for the simulation of sea ice in the PlioMIP2 models."

Additionally, the reviewer is correct about that the coarse resolution of the climate models and the location(s) of sea ice proxies on the maximum monthly sea ice extent boundary.

In the sentence "The majority of the models simulate a maximum SIE that extends, or nearly extends, into the Fram Strait and Iceland Sea Figure 10b) in at least one month (in winter) per year (Fig. 10b)," the part ", or nearly extends," was included in the paper to allow for some room for error spatially. No change was made here.

The following paragraph describes the models that match the proxy evidence completely, and does not allow for this room for error. Many models nearly match the reconstructions, and others just barely match them, and changing the definition for sea ice from a SIC of 15% to, for example, 10% would already give substantially different

results. This indicates that it is too arbitrary to conclude whether a model completely agrees or completely disagrees with a specific reconstruction. Hence, the paragraph was removed.

Review comment: Lines 382-390: the authors should more clearly state what the differences are between the paleo and future simulations. Both are forced with greenhouse-gas concentration changes, but the paleo runs are further forced by changes in the icesheets, vegetation, gate-ways? As for the changes in ice-sheets, vegetation and also the AMOC, one could argue that these simulation give a true long-term equilibrium response to greenhouse-gas changes. This is not the case for the impact of changing the Arctic gate-ways. Is there a way to quantify the impact of the latter as to make the comparison with future simulations more meaningful?

Reply: All major differences between the future and mid-Pliocene simulations are listed in lines 376-382.

It is an interesting idea to try to isolate the effects of orography, under the assumption that future climate will look similar to the mid-Pliocene in terms of CO2, ice sheets, and vegetation. Several papers have isolated the effects of the implementation of mid-Pliocene orography in their PlioMIP2 simulations and we have added these results to this paragraph.

Changed the paragraph to:

"Using PlioMIP2 simulations for potential lessons about future warming may be improved by isolating the effects of the changes in orograph. Similar changes in ice sheets and vegetation may occur in future equilibrium warm climates, but the changes in orography are definitively non-analogous to future warming. Several groups isolated the effects of the changed orography on global warming in PlioMIP2 simulations and found that it contributes, respectively, around 23% (IPSL6-CM6A-LR; Tan et al., 2020), 27% (COSMOS; Stepanek et al., 2020), and 41% (CCSM4-UoT; Chendan and Peltier, 2018) to the annual mean global warming in the mPWP simulations. Furthermore, this

warming was strongest in the high latitudes (Chandan and Peltier, 2018; Tan et al., 2020) indicating that the additional Arctic warming in PlioMIP2 simulations, as compared to future climate simulations, are likely partially caused by changes in orography that are non-analogous with the modern-day orography. These findings highlight the caution that has to be taken when using palaeoclimate simulations as analogues for future climate change."

Review comment: Lines 391-398: There are a number of studies discussing simulations of the impact of closing the Bering Strait on the AMOC strength, do they also show a moderate to strong increase in AMOC strength?

Reply: These studies did not fully implement the PlioMIP2 boundary conditions, and not all of them closed both the Bering Strait and the Canadian Archipelago Seaway. Otto-Bliesner et al. (2017) closed both Arctic Ocean gateways and found an increase of 4.5 Sv in the AMOC (~18% increase). We do not include this result in the paper as it does not implement the other PlioMIP2 boundary conditions, which may influence the magnitude of change. We do mention the papers, as the direction of change (increase in AMOC strength) corresponds. No changes made.

Review comment: Line 430: what is meant with an 'active NAO strength'? It appears that the models do not provide robust support of a change in NAO amplitude.

Reply: Indeed this is not clear. We refer to our earlier response to comments about this section.

Review comment: Line 433: Why are RCP6.0 and RCP8.5 simulations not used in the comparison if those provide a better comparison in terms of temperature changes?

Reply: Good point, ideally we would compare the simulations to RCP6.0 and RCP8.5, because they are more similar in terms of temperature change, but data was only available for the RCP4.5 projections. Since this section has since been removed (see earlier comments), we do not address this comment further.

[Figure]

Review comment: Line 455: What would such improvements in boundary conditions be? Don't the authors think that all changes in boundary conditions that are likely to have a significant effect are already included?

Reply: We do think that the most important changes in boundary conditions are incorporated, but there are still large uncertainties surrounding them. E.g. It is unclear whether the atmospheric CO2 concentration was actually 400ppm. Reducing these uncertainties could improve the simulations. We change the wording "enhanced boundary conditions" to "reducing uncertainties in boundary conditions". Furthermore, we suggest later in the conclusions that more sensitivity experiments could be carried out to quantify the effects of these uncertainties on the simulations. No changes were made here.

Review comment: Figure 11: limited data availability? The data between 60N and 67.5N is missing?

Reply: The IPCC (Masson-Delmotte et al., 2013 in this case) use 67.5-90N as their definition of the Arctic region and listed their data for this region. Changed the phrasing of "here 67.5-90 N, due to limited data availability" to "here 67.5-90 N, the definition used by Masson-Delmotte et al. (2013) and the area for which they listed data".

Technical comments: Review comment: Line 110: For me forcings are not part of model physics. Please clarify.

Reply: Indeed, this can be phrased better. Changed "Uncertainties in model physics include unconstrained forcings and uncertainties in model parameters" to "Uncertainties in model physics include processes that are not incorporated in the model and uncertainties in model parameters."

Review comment: Line 145: missing space

Reply: Good spot, fixed.

Review comment: Line 451: "11 out of 16", just for clarity.

Reply: Good suggestion, added "out of 16" for increased clarity.

Review comment: Table 1: It would be good to add to this table if the model is also used in CMIP5, CMIP6 or neither of those.

Reply: All models participating in PlioMIP2 are participants of CMIP6. The pre-industrial simulation is the piControl simulation of the CMIP6 DECK experiments, and PlioMIP2 is part of PMIP4 which is one of the projects of CMIP6. As CMIP6 models are generally different versions of their equivalent CMIP5 counterparts, we do not add information about the CMIP5 models, as this is not relevant for the current paper.

Review comment: Caption figure 3: shouldn't that be "compared to the annual mean in a given month"?

Reply: The figure depicts the ratio between the warming in a given month respective to the annual mean, for each model individually. Adjusted the caption to: "Ratio between the mean Arctic (a) SAT and (b) SST warming in a given month and the annual mean Arctic warming, for each model (and MMM) individually"

Review comment: Figure 6: what does the 'p' stand for?

Reply: Added "Depicted for both correlations are the correlation coefficient (R), the slope and the probability value (p) that when the variables are not related, a statistical result equal to or greater than observed would occur."

Review comment: Figure 11: What is shown for the RCP simulations, an average over year xx to yy?

Reply: Added (2081-2100 average) to the figure caption, and "end-of-century (2081-2100) average" to the text preceding the figure.

Please also note the supplement to this comment:
https://cp.copernicus.org/preprints/cp-2020-64/cp-2020-64-AC1-supplement.pdf

---

## Author Comment (AC2) · 3 Aug 2020

Thank you for your review. We have addressed each of your comments one-by-one and we feel that there were some substantial improvements following your comments. We also attached this author response in the supplement PDF file for you to refer.

Review comments: In the present manuscript, de Nooijer et al. present an analysis of Arctic climate as simulated by the coupled models ensemble from the PLIOMIP2 initiative. PLIOMIP2 focuses on the specific KM5c interval, peak of the mPWP. Notable improvements have also being done for the boundary conditions (e.g. closed Arctic gateways during this period). Models generally simulate an Arctic amplification larger than 2.5, increase in SAT and SST. Comparison with the few existing proxies suggest

that only few models of the ensemble are able to fit the warm climatic conditions of the particular KM5c interval. However, the lack of proxies prevent a more detailed comparison. An attempt is made to compare those new results to projections. Conclusion of the authors is that using the simulated mPWP KM5c is not yet informative for the future, given the current state of models and limitations of the design of the experiments and lack of proxies to validate the paleo-simulations. In general, what this phase 2 of the PLIOMIP initiative shows is that boundary conditions improvements and focus on a specific interval of the mPWP generally increase the agreement with the few existing proxies. However, the paper remains rather very elusive and not detailed too much about the causes of the simulated anomalies. In addition, there is a distinct dichotomy within the models with only few models increasing the MMM. An aspect that is really unclear throughout this manuscript is the impact of the models that do not used closed gateway in their simulations and how much this impact on the interpretation of the entire metrics presented here. In addition to closed gateways, individual model resolution might also have an impact on the representation of those gateways and this is not discuss here. The attempt made to compare with CMIP5 projections is to my opinion unsuccessful given the striking difference in gateways between the modern geography and that of the mPWP. In addition, the authors attempt to compare the mode of variability which is a non-sens here since the paleoclimatic simulations are equilibrium simulations while projections are transient short-term simulations. Authors warn about the lack of "slow-feedbacks" in the projections, but the contrary is also true, the short-term variability present some limitation in the paleoclimate runs. I do not advise to remove it. However, some improvements are needed to strengthen those parts and to make them meaningful in a way or in the other. The manuscript is written quite well (though in some places that I have indicated in my comments below, some improvements in the writing is needed to clarify). My impression is that this paper remains superficial and does not provide a real analysis of the Arctic warming. There is no real analysis of the causes/consequences of this warming (i.e. albedo, seasonal cycle in temperature, snow cover, westerlies etc.). . . Even if the number of proxies is limited,

the authors could deepen their analysis to compare the different models together to provide partial answers to some of the questions posed paper by the authors themselves within the different sub-section of the manuscript. They should also explore the dichotomy amongst the models visible in almost all the figures of this manuscript and the impact this dichotomy has on the MMM and thus the overall interpretation of the MMM. I therefore recommend moderate revisions.

Reply: A small response on the main summary of the reviewer, with regards to the paper remaining superficial: This paper mainly describes results and highlights differences between models. To investigate causes of the differences, e.g. because of albedo/seasonal cycle, we would need sensitivity experiments. This is not plausible for a multi-model analysis. We agree that the paper remained somewhat superficial, but we do not think it is possible to do deep analyses without sensitivity experiments.

Review comments: Line 68: I would remove "future" and just write "as warming in the Arctic directly affects. . .". This is because this is always true, not only for future. Or perhaps just reformulate in "as it is shown that projected Arctic warming affects. . .".

Reply: Good spot. The sentence has been adjusted to the suggested reformulation.

Review comments: Line 84: Would it be worth mentioning that the interest of the KM5c interval is because orbitals are similar to present? I think this is important and relevant to the comparison with projections.

Reply: Indeed, it is an important feature of the KM5c time slice that it has a similar-to-modern orbital forcing and we have added emphasis on this.

"Additionally, the KM5c time slice is characterized by a similar-to-modern orbital forcing (Haywood et al., 2013b; Prescott et al., 2014). These factors give lessons learned from the mPWP, and the KM5c time slice in particular, potential relevance for future climate change (Burke et al., 2018; Tierney et al., 2019), and this is one of the guiding principles of PlioMIP (Haywood et al., 2016)."

Review comment: Line 141: correct "model resultsaere calculated" in "model results are calculated"

Reply: Good spot, fixed.

Review comments: Line 196-203: I find interesting to note that most of the models simulate air and sea temperature values below the mean and that only a couple of models exhibit values much higher than the mean. It could also be worth mentioning this somewhere (though it is not a paper about individual model performances) because it also impacts on the interpretation that one does about the ensemble mean.

Reply: Indeed, good observation, a subset of the ensemble simulates much larger temperature anomalies than the rest of the ensemble. To note readers on the potential impacts this may have on the multi-model mean results we added the following:

"There is a large variation in the magnitude of the simulated Arctic SAT anomalies, with five out of sixteen models, namely CCSM4-Utrecht, CCSM4-UoT, CESM1.2, CESM2, and EC-Earth 3.3 all simulating much stronger anomalies than the rest of the ensemble. This subset of the ensemble raises the MMM substantially and this has to be taken into account when interpreting the MMM results. The MMM SAT anomaly for the PlioMIP2 ensemble excluding this subset of five models is 5.8 °C."

Additionally, we added a sentence about SST, as the same five models are seen here to raise the MMM. "Furthermore, the five models that simulated the largest Arctic SAT anomalies also simulate the largest Arctic SST anomalies."

In section 5.1 we discuss that this subset of the ensemble generally matches the SAT proxies best. No change was made here.

Review comments: Line 209: but did not you write that also the Bering Strait is closed in some of the models? We don't see a particularly large anomaly around this area.

Reply: The Bering Strait is closed in the PlioMIP2 simulations (mentioned in line 122) as a part of both the standard and enhanced boundary conditions. Indeed, the closure

of the Bering Strait did not lead to a large SAT anomaly. Upon closer inspection, the largest SAT anomalies are mostly above the Baffin Bay, rather than over the Canadian Archipelago. The first part of the paragraph has been adjusted accordingly. "The greatest MMM SAT anomalies in the Arctic are found in the regions with reduced ice sheet extent on Greenland (Haywood et al., 2016), which generally show warming of over 10 °C and even up to 20°C. Additionally, temperature anomalies of over 10 °C are simulated around the Baffin Bay"

Review comments: Line 212: and thus? What causes such an increase in the Baffin Bay? The lack of sea ice due to no arctic waters flowing through the CA? If yes, it would be good to mention.

Reply: This line was meant as a description of the results and of the figure. While it would be interesting to know the underlying mechanisms for the warming in this location, and while we do discuss a potential mechanism later in the paper (AMOC), we did not mean to describe the causes of the temperature increase in the Baffin Bay here. No changes were made.

Review comments: Line 196 - 215: How does the discrepancy in land sea mask, especially in the Bering Strait, affect the interpretation of the MMM in Figure 2? I would find very informative to indicate which models closed the Bering Strait and or the Canadian archipelago in Table 1. It seems from Figure 2b that only a few models keep the Bering Strait open. Are the models with open Bering Strait the ones with highest SST and SAT values (e.g. In Fig.1)?

Reply: Sorry for the confusion, all models have a closed Bering Strait and Canadian Archipelago as part of the PlioMIP2 boundary conditions. We added "in the mPWP simulation" to line 122 to emphasize this and to avoid future confusions for other readers. The stippling in Figure 2b has been removed as it was found to be incorrect after comments from reviewer 1 and they became redundant in the updated version. Description of stippling in Figure 2 caption has been removed.

Review comments: Lines 272-289: How much is the MMM-proxy comparison valid in the Canadian archipelago? I mean, in Figure 7 the proxies there are very closed to each others (while already slightly shifted for better understanding) and, how many grid points are there in in the simulations this area? Is the comparison here valid? Or not resolutiondependent? Same for Alaska?

Reply: Valid point. Given the coarse resolution of global climate models it could be impossible for simulations of SAT anomalies to match all five reconstructions in the Canadian Archipelago. We added the following: "It has to be noted, however, that SAT anomalies are underestimated at three other sites within the Canadian Archipelago. Given the resolution of global climate models and the close proximity of the sites, it may be impossible for simulations to match all five of these SAT estimates."

Review comments: Figure 8: Since the beginning, there are two distinct groups amongst the models and the MMM is shifted to higher value because of 7 models. This discrepancy between the two groups is very neat. Thus I really wonder what are the causes of such dichotomy and what is the impact on the interpretation of the MMM in the paper in general?

Reply: Indeed, good spot, there are two distinct groups amongst the models. We would argue, however, that the first group consists of the five previously discussed models (CCSM4-Utrecht, CCSM4-UoT, CESM1.2, CESM2, and EC-Earth 3.3) when looking at the median bias (rather than the extent of the box-whiskers). We already mention here that these are the models with the highest Arctic SAT anomalies. We added some emphasis on these five models and that it may be interesting to uncover why they are simulating distinctly larger anomalies than the other simulations.

"Future research into the underlying mechanisms for the increased Arctic warming in these five simulations, compared to the remaining eleven simulations in the ensemble, may form a way to uncover factors that contribute to improved data-model agreement."

Added the following to the eleventh line of the abstract; "although the degree of underestimation varies strongly between the simulations"

Added the following at the second line of the conclusion: "although large differences in the degree of underestimation exist between the simulations. The models that simulate the largest Arctic SAT anomalies tend to match the reconstructions better, and investigation into the mechanisms underlying the increased Arctic warming in these simulations may help uncover factors that could contribute to improved data-model agreement."

Review comments: Lines 320-321: but also models should also all use the same boundary conditions. Because if some fo the models do not close some fo the straits, or if they have no sufficient resolution to capture the width of some passages etc. . .how can we interpret the misfit between data and models correctly? I mean, as it is now, it is impossible to determine wether or not in some models the different boundary conditions or different physics affect the misfit and in which proportion. I know it is very difficult to modify the land-sea mask in coupled models and in some cases it will also require more computational resources to increase spatial resolution enough to capture the different gateways properly. However, at some points, we will need to do it to further advance those types of data-model exercises.

Reply: Sorry for the confusion. All models used the same boundary conditions, quoted from Haywood et al. (2020): "All model groups participating in PlioMIP2 were required to use standardised boundary condition data sets for the core midPliocene-eoi400 experiment". We added a sentence in the methods section to emphasize this. "All model groups incorporated the standardised set of boundary conditions from the PlioMIP2 experimental design in their simulations (Haywood et al., 2016)."

Review comments: Figure 10: yellow and white squares are reconstructions from proxies? I guess yes. . . but this is not mentioned in the caption.

Reply: This information has been added to the figure caption. "Depicted squares represent the locations of the reconstructions and their respective colour the inferred mPWP

sea ice conditions at that location."

Review comments: Figure 11: is the vertical Y scale in frame b) the same as in frame a)? In any case, please add the ticks for dSAT values on the graph for projections.

Reply: Indeed, the Y scales are the same. It is a good idea to add the ticks in b) also, for added clarity. This has been done and the figure has been updated.

Review comments: Lines 377- 381: When reading those lines, it seems that only $CO_2$ forcing matters here. But in many of your models, some gateways are closed, and as you cite Otto-Bliesner et al. (2017), this matters. . . Thus I disagree with the formulation of those sentences. Please also discuss the difference in Arctic geography and how this impact ton the comparison with the projections.

Reply: Indeed, $CO_2$ is not the only forcing that matters. The dominant mechanism of warming in both ensembles is $CO_2$ (for PlioMIP2 this can be found in papers from Tan et al. (2020), Chendan and Peltier (2018), and Stepanek et al. (2020)). We simply state here that this is the dominant mechanism of warming, but that there are additional mechanisms for warming in PlioMIP2. We discuss that this may be due to changes in Arctic ocean gateways or other changes in orography in the following sentence. No changes were made.

Review comments: Lines 396-400: Given the different boundary conditions, I find very difficult to make a direct comparison here. In most of PLIOMIP2 models, the Arctic gateways are closed and this generates a strengthening of the AMOC. While under modern geography, the Arctic gateways are open and a weakling of the AMOC is projected. You cannot compare those two situations here directly. In general, this short paragraph is not very clear. If you state more clearly at the beginning and in Table 1 that not all models prescribed closed gateway, this would definitely improve the reader understanding of the paper.

Reply: The main point of this paragraph is to show that there are differences between

the two ensembles, regardless of their cause, in AMOC strength and thus one of the mechanisms underlying Arctic warming. We mention in the previous paragraph that strengthening of the AMOC in PlioMIP2 is likely due to the closure of the Arctic ocean gateways. As the purpose of this paragraph is to highlight the difference, rather than investigate it, we did not make any change. Sorry again for the confusion that not all models have the closed gateway, they all do, and previous comments of the reviewer led us to put more emphasis on this to avoid confusion for future readers.

Review comments: Line 397: "This is consistent" To what does "this" refer to?

Reply: Indeed, we could make this more clear. We changed "this is consistent" to "The strengthening of the AMOC in the PlioMIP2 ensemble is consistent" and added a space to make it a separate paragraph. We also added "compared to the future climate ensembles" in the previous sentence for improved clarity.

Review comments: Subsection 7.3: To my opinion, it is very difficult to compare transient short-term projections variability with equilibrium climate variability of a few centuries (as just say line 440). Thus I find not very much straight forward and informative the conclusions from this comparison here.

Reply: Upon further inspection and thorough discussion, we decide to remove the section about the NAO/NAM. Based on comments of both Reviewer 1 and Reviewer 2. With the following reasons: - The results for both the PlioMIP2 and the RCP4.5 simulations are not very robust. There is a low signal-to-noise ratio. - The comparison of the PlioMIP2 and RCP4.5 simulations is significantly hindered by the different nature of the simulations: Equilibrium versus transient. As pointed out by reviewer 2. - The comparison is further hindered by the potential strong effect orography has on Arctic variability in the mPWP simulation. Hill et al. (2011) ascribed most of the change in the NAM they observed in the mid-Pliocene simulation to changes in orography. Since the changes in orography in PlioMIP2 are non-analogous with future climate change we do not feel that this comparison is useful.

We therefore remove Section 7.3, and make appropriate changes in the abstract, introduction, the start of Section 7, and the conclusions to represent this.

Review comments: Lines 427-429: This sentence is very unclear, please reformulate.

Reply: Thank you for the comment. The section in which this sentence was stated has been removed based on earlier comments.

Review comments: Lines 455 - 458: You state about the discrepancies between mPWP and projections simulations: "firstly the incomplete manifestation of slow responses in transient simulations". But not only, I would say also vice-versa: "the lack of transient variability in equilibrium climate". Then you state "secondly the observed differences in Arctic climate features between the ensembles": which ensembles are you referring too here? PLIOMIP1 versus PLIOMIP2 or PLIOMIP2 versus projections? If this is the second option, then I would say the entire sentence does not make sense because of course they are different, besides equilibrium versus transient, boundary conditions also differ. . .

Reply: Good point, there is a difference between comparing simulations of different climates, and different climates themselves. Indeed, both the nature (transient versus equilibrium) and boundary conditions of the ensembles differ. We focus on the differences in Arctic climate features we observe between the ensembles, and their implication for attempting to use mPWP simulations to learn about future climate change.

Changed the sentence to: "Lastly, we find differences in Arctic climate features between the PlioMIP2 ensemble and future climate ensembles, including the magnitude of Arctic amplification, changes in AMOC strength, and northern modes, which highlight that caution has to be taken when attempting to use simulations of the mPWP to learn about future climate change."

Please also note the supplement to this comment:
https://cp.copernicus.org/preprints/cp-2020-64/cp-2020-64-AC2-supplement.pdf